# Improving Online-to-Nonconvex Conversion for Smooth Optimization via Double Optimism

**Francisco Patitucci**[1*]  **Ruichen Jiang**[1*]  **Aryan Mokhtari**[1,2]
[1]The University of Texas at Austin   [2]Google Research
{fpatitucci,rjiang}@utexas.edu, mokhtari@austin.utexas.edu

## Abstract

A recent breakthrough in nonconvex optimization is the online-to-nonconvex conversion framework of Cutkosky et al. (2023), which reformulates the task of finding an $\varepsilon$-first-order stationary point as an online learning problem. When both the gradient and the Hessian are Lipschitz continuous, instantiating this framework with two different online learners achieves a complexity of $\mathcal{O}(\varepsilon^{-1.75} \log(1/\varepsilon))$ in the deterministic case and a complexity of $\mathcal{O}(\varepsilon^{-3.5})$ in the stochastic case. However, this approach suffers from several limitations: (i) the deterministic method relies on a complex double-loop scheme that solves a fixed-point equation to construct hint vectors for an optimistic online learner, introducing an extra logarithmic factor; (ii) the stochastic method assumes a bounded second-order moment of the stochastic gradient, which is stronger than standard variance bounds; and (iii) different online learning algorithms are used in the two settings.

In this paper, we address these issues by introducing an online optimistic gradient method based on a novel *doubly optimistic hint function*. Specifically, we use the gradient at an extrapolated point as the hint, motivated by two optimistic assumptions: that the difference between the hint and the target gradient remains near constant, and that consecutive update directions change slowly due to smoothness. Our method eliminates the need for a double loop and removes the logarithmic factor. Furthermore, by simply replacing full gradients with stochastic gradients and under the standard assumption that their variance is bounded by $\sigma^2$, we obtain a unified algorithm with complexity $\mathcal{O}(\varepsilon^{-1.75} + \sigma^2\varepsilon^{-3.5})$, smoothly interpolating between the best-known deterministic rate and the optimal stochastic rate.

## 1 Introduction

In this paper, we consider an unconstrained minimization problem

$$\min_{\mathbf{x}\in\mathbb{R}^d} F(\mathbf{x}), \tag{1}$$

where $F : \mathbb{R}^d \to \mathbb{R}$ is a smooth but possibly non-convex function bounded from below. With access to a deterministic gradient oracle, it is well known that gradient descent (GD) can find an $\varepsilon$-stationary point within $\mathcal{O}(\varepsilon^{-2})$ iterations when the gradient of $F$ is Lipschitz continuous, which is worst-case optimal among all first-order methods (Carmon et al., 2020). In the stochastic setting, where the oracle returns an unbiased noisy gradient with variance bounded by $\sigma^2$, stochastic gradient descent (SGD) achieves a complexity of $\mathcal{O}(\varepsilon^{-2} + \sigma^2\varepsilon^{-4})$ (Ghadimi & Lan, 2013), which is likewise worst-case optimal (Arjevani et al., 2023).

Interestingly, when both the gradient and the Hessian of $F$ are Lipschitz continuous, it becomes possible to outperform standard GD or SGD. In the deterministic setting, Carmon et al. (2017) proposed an accelerated gradient-based method that exploits directions of negative curvature, achieving a complexity of $\mathcal{O}(\varepsilon^{-1.75} \log(1/\varepsilon))$. Concurrently, Agarwal et al. (2017) introduced an approximate variant of the cubic regularization method (Nesterov & Polyak, 2006), which uses only first-order gradients and access to Hessian-vector products, and achieves a complexity of $\mathcal{O}(\varepsilon^{-1.75} \log(d/\varepsilon))$.

---

*Equal contribution

More recently, Li & Lin (2022; 2023) eliminated the logarithmic factor from these bounds by incorporating restarting techniques into accelerated gradient and heavy-ball methods, achieving the state-of-the-art complexity of $\mathcal{O}(\varepsilon^{-1.75})$. However, the methods proposed in (Carmon et al., 2017; Agarwal et al., 2017) are complex and rely on multiple subroutines to exploit negative curvature, while the restarting techniques in (Li & Lin, 2022; 2023) involve lengthy derivations, and their extension to the stochastic setting remains unclear. In the stochastic setting, similar improvements over SGD have been achieved using different algorithmic techniques (Allen-Zhu, 2018a;b; Tripuraneni et al., 2018; Fang et al., 2019). To our knowledge, the best known result is due to Cutkosky & Mehta (2020), who proposed normalized SGD with momentum and extrapolation, achieving $\mathcal{O}(\varepsilon^{-2} + \sigma^2 \varepsilon^{-3.5})$. While this method attains the optimal rate of $\mathcal{O}(\sigma^2 \varepsilon^{-3.5})$ for the stochastic setting (Arjevani et al., 2020), it fails to achieve the accelerated rate of $\mathcal{O}(\varepsilon^{-1.75})$ for the deterministic setting. Notably, to the best of our knowledge, no single unified algorithm simultaneously attains the best-known guarantees in both settings.

To address this gap, a promising step toward unification is the *online-to-nonconvex* (O2NC) conversion proposed in (Cutkosky et al., 2023), which offers a general and powerful framework for nonconvex optimization across various settings. The core idea is to reduce the problem of finding a stationary point of a nonconvex function to an online learning problem over the update directions. This reduction enables converting an online learning algorithm into a nonconvex optimization method, which greatly simplifies convergence analysis. In the smooth setting considered in this paper, by instantiating two different online learning algorithms, Cutkosky et al. (2023) establish a complexity of $\mathcal{O}(\varepsilon^{-1.75} \log(1/\varepsilon))$ in the deterministic case and $\mathcal{O}(G^2 \varepsilon^{-3.5})$ in the stochastic case, where $G^2$ is an upper bound on the second-order moment of the stochastic gradient. Despite its conceptual elegance, the O2NC framework has several limitations. First, in the deterministic setting, the algorithm relies on a fixed-point iteration as a subroutine, introducing an extra logarithmic factor in the convergence rate and adding complexity to the algorithmic design. Second, in the stochastic setting, it requires a stronger assumption that the second-order moment of the stochastic gradient is uniformly bounded, which can be restrictive in practice. Third, the framework requires different online learning algorithms for the stochastic and deterministic settings, limiting its modularity and complicating potential extensions.

**Our contribution.** Building on the O2NC framework, we propose a simple yet effective algorithm that replaces the fixed-point iteration subroutine in (Cutkosky et al., 2023) with a single stochastic gradient call at an extrapolated point, achieving a complexity of $\mathcal{O}(\varepsilon^{-1.75} + \sigma^2 \varepsilon^{-3.5})$. This eliminates the additional logarithmic factor and smoothly interpolates between the best-known deterministic and stochastic rates. Our core idea is a novel *doubly optimistic* online gradient method, which aims to predict the gradient using a carefully crafted hint function. Inspired by standard optimistic methods (Chiang et al., 2012; Rakhlin & Sridharan, 2013; Joulani et al., 2020), our first optimistic assumption is that the discrepancy between the gradient and the hint evolves slowly over time. Crucially, we introduce a second optimism that the update direction itself also evolves slowly, which we exploit using the function's smoothness to guide the design of the hint. To the best of our knowledge, this is the first algorithm to seamlessly attain the best-known guarantees in both deterministic and stochastic settings, resolving a key limitation of the original O2NC framework.

## 1.1 Additional related work

**Deterministic setting** Beyond the works reviewed in the introduction, Marumo & Takeda (2024a;b) built on the restarting technique from (Li & Lin, 2022; 2023) to develop parameter-free algorithms using line search, thereby removing the need for prior knowledge of problem-specific constants. Complementing these upper bounds, Carmon et al. (2021) established a lower bound of $\Omega(\varepsilon^{-\frac{12}{7}})$ for functions with Lipschitz gradient and Hessian. This leaves a gap of $\mathcal{O}(\varepsilon^{-\frac{1}{28}})$ between the best-known upper and lower bounds in this setting. Several works also aimed to find second-order stationary points (Agarwal et al., 2017; Carmon et al., 2018; Jin et al., 2018; Allen-Zhu & Li, 2018; Xu et al., 2017; Royer & Wright, 2018; Royer et al., 2020) using gradient or Hessian-vector product oracles, but their complexity remains no better than $\mathcal{O}(\varepsilon^{-1.75})$.

**Stochastic setting** Several works have shown improved guarantees by either strengthening the oracle or imposing additional restrictions on the noise. Specifically, with a stronger oracle that has access to Hessian-vector products, Arjevani et al. (2020) proposed a variance-reduction algorithm

that combines stochastic gradients with Hessian-vector products, attaining the optimal complexity of $\mathcal{O}(\sigma^2 \varepsilon^{-3})$. Moreover, another line of work relies on the mean-squared smoothness assumption on stochastic gradients, i.e., $\mathbb{E}[\|\nabla f(\mathbf{x}; \xi) - \nabla f(\mathbf{y}; \xi)\|^2] \leq \bar{L}^2 \|\mathbf{x} - \mathbf{y}\|^2$ for all $\mathbf{x}$ and $\mathbf{y}$ with some constant $\bar{L}$; under this condition, variance-reduction-based methods such as SPIDER (Fang et al., 2018) and SNVRG (Zhou et al., 2020) achieved a complexity of $\mathcal{O}(\sigma \varepsilon^{-3})$. Under a slightly stricter assumption that $\nabla f(\mathbf{x}; \xi)$ is Lipschitz continuous with probability one, Cutkosky & Orabona (2019) established the complexity of $\mathcal{O}(\varepsilon^{-2} + \sigma \varepsilon^{-3})$. It is also worth mentioning that the above methods require a multi-point stochastic oracle, which computes the stochastic gradients at two different points with the same random seed $\xi$. Finally, the lower bounds in (Arjevani et al., 2023) established that these complexity results are worst-case optimal.

**Online-to-nonconvex conversion**   Apart from the smooth nonconvex optimization considered in this paper, the original O2NC conversion by Cutkosky et al. (2023) also addressed the complexity of nonsmooth nonconvex optimization first considered in Zhang et al. (2020), yielding an optimal stochastic algorithm. The framework has been extended and refined in several follow-up works (Zhang & Cutkosky, 2024; Ahn et al., 2024a), and it has been shown that various practical algorithms, such as SGD with momentum and Adam, can be interpreted within the O2NC framework (Zhang & Cutkosky, 2024; Ahn et al., 2024b; Ahn & Cutkosky, 2024; Ahn et al., 2024a).

**Comparison with Jiang et al. (2025)**   A recent work by Jiang et al. (2025) also built upon the online-to-non-convex conversion framework introduced by Cutkosky et al. (2023), leveraging optimistic online gradient descent to tackle the resulting online learning problem. However, their focus is on quasi-Newton methods in the deterministic setting, whereas our work focuses on first-order methods in both stochastic and deterministic settings. While that approach shares a similar high-level structure and relies on common core lemmas as ours, they differ fundamentally in their hint constructions. Specifically, in our paper, our hint is based on the stochastic gradient at an extrapolated point, whereas Jiang et al. (2025) employed a second-order approximation, where the approximate Hessian matrix is updated via an online learning algorithm in the space of matrices. Due to the different hint constructions, our analysis exploits negative terms in the regret of optimistic online gradient descent to obtain tighter regret bounds—terms that are neglected in (Jiang et al., 2025).

## 2 PRELIMINARIES AND BACKGROUND

Throughout the paper, we assume that $F$ satisfies two key assumptions described below. Unless otherwise specified, we use $\|\cdot\|$ to denote the $\ell_2$-norm for vectors and the operator norm of matrices.

**Assumption 2.1.** *We have $\|\nabla F(\mathbf{x}) - \nabla F(\mathbf{y})\| \leq L_1 \|\mathbf{x} - \mathbf{y}\|$ for any $\mathbf{x}, \mathbf{y} \in \mathbb{R}^d$.*

**Assumption 2.2.** *We have $\|\nabla^2 F(\mathbf{x}) - \nabla^2 F(\mathbf{y})\| \leq L_2 \|\mathbf{x} - \mathbf{y}\|$ for any $\mathbf{x}, \mathbf{y} \in \mathbb{R}^d$.*

In the stochastic setting, we assume access to an unbiased gradient oracle with bounded variance.

**Assumption 2.3.** *We assume that we have access to a stochastic gradient oracle satisfying*

$$\mathbb{E}_{\xi \sim \mathcal{D}}[\nabla f(\mathbf{x}; \xi)] = \nabla F(\mathbf{x}) \quad and \quad \mathbb{E}_{\xi \sim \mathcal{D}} \|\nabla f(\mathbf{x}; \xi) - \nabla F(\mathbf{x})\|^2 \leq \sigma^2, \quad \forall \mathbf{x} \in \mathbb{R}^d, \quad (2)$$

*where $\mathcal{D}$ is a data distribution.*

### 2.1 ONLINE-TO-NONCONVEX CONVERSION

As discussed, the online-to-nonconvex conversion (O2NC) framework proposed in Cutkosky et al. (2023) provides a unified approach that reformulates the task of finding a stationary point of a nonconvex function $F$ as an online convex optimization (OCO) problem. By instantiating this framework with different online learning algorithms to solve the resulting OCO problem, one can derive various algorithms for nonconvex optimization. In this section, we first recap the core ideas behind the O2NC framework.

Consider the general update rule $\mathbf{x}_n = \mathbf{x}_{n-1} + \boldsymbol{\Delta}_n$, where we assume that the update direction $\boldsymbol{\Delta}_n$ has a bounded norm, i.e., $\|\boldsymbol{\Delta}_n\| \leq D$ for a constant $D > 0$. The starting point of Cutkosky et al. (2023) is to apply the fundamental theorem of calculus to characterize the function decrease between two consecutive points as

$$F(\mathbf{x}_n) - F(\mathbf{x}_{n-1}) = \langle \boldsymbol{\nabla}_n, \mathbf{x}_n - \mathbf{x}_{n-1} \rangle = \langle \boldsymbol{\nabla}_n, \boldsymbol{\Delta}_n \rangle, \quad (3)$$

where in the above expression $\boldsymbol{\nabla}_n = \int_0^1 \nabla F(\mathbf{x}_{n-1} + s(\mathbf{x}_n - \mathbf{x}_{n-1}))\, ds$ is the average gradient along the line segment between $\mathbf{x}_{n-1}$ and $\mathbf{x}_n$. Given the equality in (3) and upper bound on the norm of $\boldsymbol{\Delta}_n$, the ideal choice of $\boldsymbol{\Delta}_n$ to maximize the function value decrease is to set it as $\boldsymbol{\Delta}_n = -D\frac{\boldsymbol{\nabla}_n}{\|\boldsymbol{\nabla}_n\|}$, which leads to a function decrease of $-D\|\boldsymbol{\nabla}_n\|$. However, the implementation of this update poses several challenges, which we outline below along with potential solutions.

First, computing $\boldsymbol{\nabla}_n$ involves evaluating an integral over a segment, which is computationally expensive. One remedy is to approximate the average gradient over the segment by evaluating the gradient at a randomly selected point along the line segment connecting $\mathbf{x}_{n-1}$ and $\mathbf{x}_n$; in expectation, this yields the same value as $\boldsymbol{\nabla}_n$. An alternative, deterministic approach is to use the gradient at the midpoint $\frac{1}{2}(\mathbf{x}_{n-1} + \mathbf{x}_n)$ as an approximation to $\boldsymbol{\nabla}_n$. Under the assumption of Lipschitz continuity of the Hessian, it can be shown that the resulting error is $\mathcal{O}(LD^2)$. In this work, we adopt the second approach and define the midpoint as $\mathbf{w}_n = \frac{1}{2}(\mathbf{x}_{n-1} + \mathbf{x}_n)$.

However, two key challenges remain. First, in the stochastic setting, we do not have access to the gradient $\nabla F(\mathbf{w}_n)$, but only to a stochastic gradient oracle (see Assumption 2.3). To resolve this, we use an unbiased stochastic estimate of the gradient at the midpoint, denoted by $\mathbf{g}_n = \nabla f(\mathbf{w}_n; \xi_n)$, where $\xi_n \sim \mathcal{D}$. In the rest of the paper, we assume access to a stochastic gradient oracle with variance $\sigma^2$. By setting $\sigma^2 = 0$, our analysis and algorithm naturally recover the deterministic setting.

The second and perhaps most critical challenge, which arises in both deterministic and stochastic settings, is that computing the (stochastic) gradient at the midpoint $\mathbf{w}_n$ requires knowledge of the next iterate $\mathbf{x}_n$, which itself depends on the update direction $\boldsymbol{\Delta}_n$. Since $\mathbf{x}_n$ is not available when choosing $\boldsymbol{\Delta}_n$, this creates a circular dependency. To overcome this, we cast the selection of $\boldsymbol{\Delta}_n$ as an *online learning problem*, where $\boldsymbol{\Delta}_n$ is the action taken at round $n$, and the loss is defined as $\langle \mathbf{g}_n, \boldsymbol{\Delta}_n \rangle$. This loss reflects how well the chosen direction aligns with the negative of the (stochastic) gradient evaluated at the midpoint, thus guiding the updates even in the absence of $\mathbf{x}_n$ at decision time. Concretely, the online-to-nonconvex conversion framework proceeds as follows

- Update $\mathbf{x}_n = \mathbf{x}_{n-1} + \boldsymbol{\Delta}_n$ and $\mathbf{w}_n = \mathbf{x}_{n-1} + \frac{1}{2}\boldsymbol{\Delta}_n$;
- Construct a stochastic estimate $\mathbf{g}_n = \nabla f(\mathbf{w}_n; \xi_n)$ with $\xi_n \sim \mathcal{D}$;
- Feed the loss $\langle \mathbf{g}_n, \boldsymbol{\Delta} \rangle$ to an online learning algorithm to obtain $\boldsymbol{\Delta}_{n+1}$ with $\|\boldsymbol{\Delta}_{n+1}\| \leq D$.

A key result from Cutkosky et al. (2023) establishes that convergence to stationarity in Problem (1) can be related to a specific notion of regret in the online learning problem above. Specifically, suppose the process runs for $M = KT$ iterations, divided into $K$ episodes of length $T$ each. We then define the $K$-*shifting regret* as

$$\text{Reg}_T(\mathbf{u}^1, \ldots, \mathbf{u}^K) = \sum_{k=1}^{K} \sum_{n=(k-1)T+1}^{kT} \langle \mathbf{g}_n, \boldsymbol{\Delta}_n - \mathbf{u}^k \rangle, \tag{4}$$

where $\{\mathbf{u}^k\}_{k=1}^{K}$ is a sequence of comparator vectors to be specified. In the next proposition, we bound the gradient norm at the average iterates in terms of $\text{Reg}_T(\mathbf{u}^1, \ldots, \mathbf{u}^K)$. Due to space constraints, we defer the detailed discussions and the proof to the Appendix.

**Proposition 2.1.** *Suppose that Assumptions 2.2 and 2.3 hold. For $k = 1, \ldots, K$, define $\bar{\mathbf{w}}^k = \frac{1}{T} \sum_{n=(k-1)T+1}^{kT} \mathbf{w}_n$ and $\mathbf{u}^k = -D\frac{\sum_{n=(k-1)T+1}^{kT} \mathbf{g}_n}{\|\sum_{n=(k-1)T+1}^{kT} \mathbf{g}_n\|}$. Recall the definition of $\text{Reg}_T(\mathbf{u}^1, \ldots, \mathbf{u}^K)$ in (4). Then we have*

$$\mathbb{E}\left[\frac{1}{K} \sum_{k=1}^{K} \|\nabla F(\bar{\mathbf{w}}^k)\|\right] \leq \frac{F(\mathbf{x}_0) - F^*}{DKT} + \frac{\mathbb{E}\left[\text{Reg}_T(\mathbf{u}^1, \ldots, \mathbf{u}^K)\right]}{DKT} + \frac{L_2}{48}D^2 + \frac{L_2}{2}T^2 D^2 + \frac{\sigma}{\sqrt{T}}.$$

In Proposition 2.1, the returned points $\{\bar{\mathbf{w}}^k\}_{k=1}^{K}$ represent the average iterates within each episode, and the comparators $\{\mathbf{u}^k\}_{k=1}^{K}$ are defined based on the sum of stochastic gradients in each episode. Moreover, the proposition shows that the average gradient norm at $\{\bar{\mathbf{w}}^k\}_{k=1}^{K}$ can be bounded in terms of the $K$-shifting regret with respect to these comparators. Hence, to obtain a gradient complexity bound for finding a stationary point of the function $F$, it suffices to use an online learning algorithm

to minimize the regret. This approach yields an explicit update for $\boldsymbol{\Delta}_n$, and consequently for the iterates. Specifically, the online learning problem we need to solve is presented in the box below.

In Section 3, we introduce our *online doubly optimistic gradient method* along with a new hint function, designed to efficiently solve the above online learning problem in both deterministic and stochastic settings. Before that, we briefly review the online learning algorithms proposed in Cutkosky et al. (2023), originally developed for these settings, and highlight their limitations. Notably, the approaches differ between the deterministic and stochastic cases to achieve the best possible regret bounds and corresponding gradient complexity.

---

**Online Learning Problem**

For $n = 1, \ldots, KT$:

- The learner chooses $\boldsymbol{\Delta}_n \in \mathbb{R}^d$ such that $\|\boldsymbol{\Delta}_n\| \leq D$;
- Sample $\xi_n \sim \mathcal{D}$ and compute $\mathbf{g}_n = \nabla f(\frac{1}{2}(\mathbf{x}_n + \mathbf{x}_{n-1}); \xi_n)$, where $\mathbf{x}_n = \mathbf{x}_{n-1} + \boldsymbol{\Delta}_n$;
- The learner observes the loss $\ell_n(\boldsymbol{\Delta}_n) = \langle \mathbf{g}_n, \boldsymbol{\Delta}_n \rangle$.

**Goal:** Minimize the regret $\text{Reg}_T(\mathbf{u}^1, \ldots, \mathbf{u}^K) = \sum_{k=1}^{K} \sum_{n=(k-1)T+1}^{kT} \langle \mathbf{g}_n, \boldsymbol{\Delta}_n - \mathbf{u}^k \rangle$.

---

**Deterministic setting.**  In the deterministic setting, Cutkosky et al. (2023) employs an optimistic gradient method, which relies on a hint vector $\mathbf{h}_n$ to approximate the true gradient $\mathbf{g}_n$ as closely as possible. The regret of these methods is governed by the approximation error between $\mathbf{h}_n$ and $\mathbf{g}_n$, as shown later in Lemma 3.1. Hence, a natural choice for $\mathbf{h}_n$ is to leverage the smoothness of $F$ and set $\mathbf{h}_n = \mathbf{g}_{n-1}$. However, this only leads to a complexity of $\mathcal{O}(\varepsilon^{-\frac{11}{6}})$ that is suboptimal compared to the state-of-the-art, as discussed in Section 3.2. To achieve a better rate, Cutkosky et al. (2023) build on a variant of optimistic gradient descent from Chen et al. (2021) and propose a more sophisticated hint construction. Specifically, starting from $\hat{\boldsymbol{\Delta}} = 0$, they follow the update rule $\boldsymbol{\Delta}_n = \Pi_{\{\|\boldsymbol{\Delta}\| \leq D\}}(\hat{\boldsymbol{\Delta}}_n - \eta \mathbf{h}_n)$, $\quad \hat{\boldsymbol{\Delta}}_{n+1} = \Pi_{\{\|\boldsymbol{\Delta}\| \leq D\}}(\hat{\boldsymbol{\Delta}}_n - \eta \mathbf{g}_n)$ where $\Pi_{\|\boldsymbol{\Delta}\| \leq D}(\cdot)$ denotes projection onto the Euclidean ball of radius $D$. As shown by the update rule, the algorithm maintains two sequences: $\hat{\boldsymbol{\Delta}}_n$ and $\boldsymbol{\Delta}_n$. The auxiliary sequence $\hat{\boldsymbol{\Delta}}_n$ is designed to closely approximate $\boldsymbol{\Delta}_n$ and plays a key role in constructing the hint vector $\mathbf{h}_n$. Specifically, $\mathbf{h}_n$ is computed through an inner loop that solves the following fixed-point equation for $\mathbf{h}$, which depends on $\hat{\boldsymbol{\Delta}}_n$, $\mathbf{h} = \nabla F\left(\mathbf{x}_{n-1} + \frac{1}{2}\Pi_{\{\|\boldsymbol{\Delta}\| \leq D\}}[\hat{\boldsymbol{\Delta}}_n - \eta \mathbf{h}]\right)$. While successfully achieving the complexity of $\mathcal{O}(\varepsilon^{-1.75} \log(\frac{1}{\varepsilon}))$, this hint construction introduces two drawbacks: the inner loop adds an extra logarithmic factor to the final complexity bound, and the fixed-point formulation complicates its extension to the stochastic setting.

**Stochastic setting.**  In the stochastic setting, Cutkosky et al. (2023) proposes using standard projected online gradient descent (OGD) to solve the online learning problems. The update rule follows $\boldsymbol{\Delta}_{n+1} = \Pi_{\{\|\boldsymbol{\Delta}\| \leq D\}}(\boldsymbol{\Delta}_n - \eta \mathbf{g}_n)$. Under the assumption that the stochastic gradient has a bounded second moment, i.e., $\mathbb{E}[\|\nabla f(\mathbf{x}; \xi)\|^2] \leq G^2$ for all $\mathbf{x}$, they prove that the regret of OGD can be bounded by $\text{Reg}_T(\mathbf{u}^1, \ldots, \mathbf{u}^K) = \mathcal{O}(KGD\sqrt{T})$. Under Assumption 2.2 and with proper tuning of the hyperparameters, this leads to an improved complexity of $\mathcal{O}(\varepsilon^{-3.5})$. However, note that this rate does not improve in the deterministic setting, where $\sigma = 0$. Moreover, the bounded second moment condition required to guarantee regret bounds using OGD can be a restrictive assumption to impose, which is not necessary in other existing methods Cutkosky & Mehta (2020).

To address the limitations identified above in both the stochastic and deterministic settings, we develop our "Online Doubly Optimistic Gradient" method. In the following section, we explain how the algorithm leverages two layers of optimism to overcome these challenges and provide a comprehensive analysis of its complexity.

## 3 PROPOSED ALGORITHM

This section introduces our proposed algorithm and the key ideas behind its design. Building on the O2NC framework Cutkosky et al. (2023), we reformulate the problem of finding a stationary point of

$F$ as an online learning task. To solve the resulting OCO problem, we present a modified variant of the Online Optimistic Gradient method, distinct from the one discussed in the previous section. In addition, as our main algorithmic contribution, we propose a novel hint function derived from a two-level optimism scheme, detailed below. Hence, we refer to our approach as the "Online Doubly Optimistic Gradient" method.

## 3.1 ONLINE OPTIMISTIC GRADIENT METHOD

As mentioned earlier, we also aim to solve the online learning problem described in Section 2.1. Unlike the optimistic template used in Cutkosky et al. (2023), which requires two projections per iteration, we adopt a simpler variant of the optimistic gradient method that requires only a single projection. Specifically, for $n > 1$, we update the direction $\boldsymbol{\Delta}_{n+1}$ using the following rule

$$\boldsymbol{\Delta}_{n+1} = \Pi_{\|\boldsymbol{\Delta}\| \leq D} \left( \boldsymbol{\Delta}_n - \eta \mathbf{h}_{n+1} - \eta(\mathbf{g}_n - \mathbf{h}_n) \right), \quad \forall n > 1, \tag{5}$$

and for $n = 1$, we initialize with $\boldsymbol{\Delta}_1 = \arg\min_{\|\boldsymbol{\Delta}\| \leq D} \langle \mathbf{h}_1, \boldsymbol{\Delta} \rangle$.

To motivate the update rule introduced above, observe that a natural approach would be to base the update of $\boldsymbol{\Delta}_{n+1}$ on the current gradient $\mathbf{g}_{n+1}$, but this is not feasible because $\mathbf{g}_{n+1}$ is revealed only after $\boldsymbol{\Delta}_{n+1}$ has already been selected. To address this challenge, we leverage online optimistic learning techniques from Rakhlin & Sridharan (2013); Joulani et al. (2020), introducing the first layer of optimism in our *doubly optimistic algorithm*. These methods rely on a hint vector $\mathbf{h}_n$ that approximates $\mathbf{g}_n$ as accurately as possible. The intuition of this approach assumes that the deviation between the true gradient and the hint remains relatively stable across iterations, i.e., $\mathbf{g}_{n+1} - \mathbf{h}_{n+1} \approx \mathbf{g}_n - \mathbf{h}_n$. Accordingly, one can estimate $\mathbf{g}_{n+1} \approx \mathbf{h}_{n+1} + \mathbf{g}_n - \mathbf{h}_n$, and the update combines an optimistic prediction using $\mathbf{h}_{n+1}$ with a correction based on the observed error $\mathbf{g}_n - \mathbf{h}_n$.

Next, we present the $K$-shifting regret bound for the optimistic gradient method described in (5). This result holds for any hint function and, accordingly, depends on the approximation error between $\mathbf{g}_n$ and $\mathbf{h}_n$, as shown in the following lemma.

**Lemma 3.1.** *Consider executing the update described in* (5) *with a constant step size $\eta$. Then the $K$-shifting regret can be bounded by*

$$\mathrm{Reg}_T(\mathbf{u}^1, \ldots, \mathbf{u}^K) \leq \frac{4KD^2}{\eta} + \frac{5\eta}{2} \sum_{n=1}^{KT} \|\mathbf{g}_n - \mathbf{h}_n\|^2 - \frac{1}{4\eta} \sum_{n=2}^{KT} \|\boldsymbol{\Delta}_n - \boldsymbol{\Delta}_{n-1}\|^2$$

.

Lemma 3.1 highlights the crucial role of the hint vector $\mathbf{h}_n$: the more accurately it approximates the true gradient $\mathbf{g}_n$, the more favorable the convergence behavior of the algorithm becomes. Precise hints tighten the regret bound, while inaccurate approximations can substantially degrade performance. Lemma 3.1 also establishes the foundation for the second layer of optimism in our algorithm. In particular, it features a negative term, $\|\boldsymbol{\Delta}_n - \boldsymbol{\Delta}_{n-1}\|^2$, which was previously neglected in Cutkosky et al. (2023). Under the smoothness assumption, we can reasonably expect that $\boldsymbol{\Delta}_n \approx \boldsymbol{\Delta}_{n-1}$. Our second layer of optimism capitalizes on this assumption and incorporates the previously neglected term, establishing a connection between this term and $\|\mathbf{g}_n - \mathbf{h}_n\|^2$ under the Lipschitz continuity of the gradient. This connection allows us to use telescoping techniques, thereby improving upon previous convergence rates within this framework. We now address the critical question of how to construct the hint sequence $\mathbf{h}_n$ to minimize the regret bound established in Lemma 3.1.

## 3.2 HINT CONSTRUCTION

For ease of exposition, we first assume access to a deterministic gradient oracle; we will return to the stochastic setting at the end of this section. As explained in Section 3.1, our primary objective in hint construction is to estimate $\mathbf{g}_n = \nabla F(\mathbf{w}_n)$ with maximum accuracy using only information available up to $\mathbf{x}_{n-1}$, as illustrated in Figure 1. Leveraging the function's smoothness properties, an intuitive first approximation is to reuse the previous gradient estimate, assuming $\mathbf{g}_n \approx \mathbf{g}_{n-1}$—effectively positing that the gradient at $\mathbf{w}_n$ approximates the gradient at $\mathbf{w}_{n-1}$. This leads to the following update rule, which corresponds to the standard optimistic online gradient descent

$$\boldsymbol{\Delta}_{n+1} = \Pi_{\|\boldsymbol{\Delta}\| \leq D} \left( \boldsymbol{\Delta}_n - \eta(2\mathbf{g}_n - \mathbf{g}_{n-1}) \right).$$

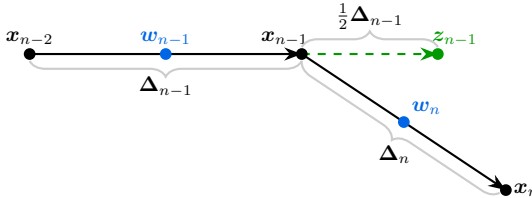

Figure 1: Illustration for the extrapolated point in our algorithm.

However, this straightforward approximation fails to exploit the negative term $\|\mathbf{\Delta}_n - \mathbf{\Delta}_{n-1}\|^2$ in Lemma 3.1. With this approach, $\|\mathbf{g}_n - \mathbf{h}_n\| \leq L\|\mathbf{w}_n - \mathbf{w}_{n-1}\|$ becomes proportional to $\|\mathbf{\Delta}_n + \mathbf{\Delta}_{n-1}\|$, preventing the application of telescoping arguments. Consequently, this method yields an overall gradient complexity of $\mathcal{O}(\varepsilon^{-11/6})$ in the deterministic setting (Cutkosky et al., 2023), which is worse than the best-known bound of $\mathcal{O}(\varepsilon^{-7/4})$.

Cutkosky et al. (2023) improved this rate under the O2NC framework by constructing the hint $\mathbf{h}_n$ through a fixed-point iteration scheme. Their approach develops an approximation $\hat{\mathbf{\Delta}}_n$ of $\mathbf{\Delta}_n$ via multiple iteration steps, estimating $\mathbf{g}_n \approx \nabla F(\mathbf{x}_{n-1} + \frac{1}{2}\hat{\mathbf{\Delta}}_n)$. While this method achieves an improved convergence rate of $\mathcal{O}(\varepsilon^{-7/4}\log(1/\varepsilon))$ in the deterministic setting, it introduces an additional logarithmic factor. Moreover, this construction is complex and presents challenges when extended to stochastic environments. These limitations motivate our search for an alternative that maintains theoretical guarantees while offering greater simplicity and broader applicability.

Our method not only removes the logarithmic factor in the deterministic case but also matches the known lower bound in the stochastic case, achieving a complexity of $\mathcal{O}(\varepsilon^{-1.75} + \sigma^2\varepsilon^{-3.5})$. We leverage our second layer of optimism to improve upon previous results. Referring to Figure 1, our goal is to estimate the gradient at $\mathbf{w}_n$ using only information available up to $\mathbf{x}_{n-1}$. Since $\mathbf{w}_n = \mathbf{x}_{n-1} + \frac{1}{2}\mathbf{\Delta}_n$ and we can reasonably approximate $\mathbf{\Delta}_n \approx \mathbf{\Delta}_{n-1}$ in the smooth setting, we propose to estimate the gradient at $\mathbf{w}_n$ using the gradient at an extrapolated point $\mathbf{z}_{n-1} = \mathbf{x}_{n-1} + \frac{1}{2}\mathbf{\Delta}_{n-1}$. This leads to the following elegant construction for the hint

$$\mathbf{h}_n = \nabla F(\mathbf{z}_{n-1}) = \nabla F(\mathbf{x}_{n-1} + \frac{1}{2}\mathbf{\Delta}_{n-1}). \tag{6}$$

We use the convention $\mathbf{\Delta}_0 = 0$, which implies that $\mathbf{h}_1 = \nabla F(\mathbf{x}_0)$. This approximation has a clear mathematical justification from the regret analysis. Recalling from Lemma 3.1 the negative term $\|\mathbf{\Delta}_n - \mathbf{\Delta}_{n-1}\|^2$ that was previously neglected in Cutkosky et al. (2023), we can now see that under this hint construction and using Assumption 2.1, we have $\|\mathbf{g}_n - \mathbf{h}_n\| \propto \|\mathbf{w}_n - \mathbf{z}_{n-1}\|$, and by construction $\|\mathbf{w}_n - \mathbf{z}_{n-1}\| = \frac{1}{2}\|\mathbf{\Delta}_n - \mathbf{\Delta}_{n-1}\|$. With appropriate stepsize selection, this allows us to leverage telescoping between these two terms and obtain tighter convergence rates.

The novelty of our hint construction lies in its simplicity: it requires only one gradient call compared to the multi-step procedure in Cutkosky et al. (2023), and it significantly simplifies the analysis. In particular, this makes it especially well-suited for the stochastic setting, where the only modification needed is replacing the deterministic gradient with its stochastic counterpart, i.e., $\mathbf{h}_{n+1} = \nabla f(\mathbf{z}_n; \xi_n)$, where $\xi_n$ is the random sample drawn at iteration $n$. In this setting, we continue to leverage the telescoping argument, with only an additional term accounting for the stochastic gradient variance. The regret incurred by our hint is formalized in the following lemma.

**Lemma 3.2.** *Consider executing the update described in* (5) *with the hint construction defined in* (6) *(or its stochastic counterpart in the stochastic setting), using step size $\eta \leq \frac{1}{\sqrt{3}L_1}$. Then we can bound one episode of the $K$-shifting regret, as*

$$\mathbb{E}\left[\sum_{n=(k-1)T+1}^{kT} \langle \mathbf{g}_n, \mathbf{\Delta}_n - \mathbf{u}^k\rangle\right] \leq \frac{4D^2}{\eta} + \frac{9L_1^2\eta D^2}{2} + 6\eta T\sigma^2.$$

The stochastic variant of our proposed method is summarized in Algorithm 1. The deterministic version can be obtained by simply replacing the stochastic gradients with their deterministic coun-

---

**Algorithm 1** Online Doubly Optimistic Gradient Algorithm

---

**Require:** Initial point $\mathbf{x}_0$, $K, T \in \mathbb{N}$, radius $D$
**Initialize:** $\boldsymbol{\Delta}_1 = -D\frac{\nabla f(\mathbf{x}_0;\xi_0)}{\|\nabla f(\mathbf{x}_0;\xi_0)\|}$, $\mathbf{h}_1 = \nabla f(\mathbf{x}_0;\xi_0)$
 1: **for** $n = 1$ to $KT$ **do**
 2:    Set $\mathbf{x}_n = \mathbf{x}_{n-1} + \boldsymbol{\Delta}_n$, $\mathbf{w}_n = \mathbf{x}_{n-1} + \frac{1}{2}\boldsymbol{\Delta}_n$, $\mathbf{z}_n = \mathbf{x}_n + \frac{1}{2}\boldsymbol{\Delta}_n$
 3:    Set $\boldsymbol{\Delta}_{n+1} = \Pi_{\|\boldsymbol{\Delta}\|\leq D}\left(\boldsymbol{\Delta}_n - \eta_n\nabla f(\mathbf{z}_n;\xi_n) - \eta_n(\nabla f(\mathbf{w}_n;\xi_n) - \nabla f(\mathbf{z}_{n-1};\xi_{n-1}))\right)$
 4: **end for**
 5: Set $\mathbf{w}_t^k = \mathbf{w}_{(k-1)T+t}$ for $k = 1, \ldots, K$ and $t = 1, \ldots, T$
 6: Set $\bar{\mathbf{w}}^k = \frac{1}{T}\sum_{t=1}^T \mathbf{w}_t^k$ for $k = 1, \ldots, K$
 7: **return** $\hat{\mathbf{w}} = \begin{cases} \text{Deterministic Oracle: } \arg\min_{\bar{\mathbf{w}}\in\{\bar{\mathbf{w}}^1,\ldots,\bar{\mathbf{w}}^K\}} \|\nabla F(\bar{\mathbf{w}})\|, \\ \text{Stochastic Oracle: Uniformly sample } \{\bar{\mathbf{w}}^1, \ldots, \bar{\mathbf{w}}^K\}. \end{cases}$

---

terparts. In the following section, we present a comprehensive complexity analysis to establish the convergence rate of the algorithm.

### 3.3 COMPLEXITY ANALYSIS

Thus far, we have presented our proposed "Online Doubly Optimistic Gradient" method in Algorithm 1, and discussed the construction of the hint vector $\mathbf{h}_n$. Next, leveraging Lemma 3.2 and Proposition 2.1, we establish the convergence rate of our proposed method.

**Theorem 3.3.** *If Assumptions 2.1–2.3 hold and we set* $D = \Theta\left(\min\left\{\left(ML_2^{\frac{1}{5}}\sigma^{\frac{4}{5}}\right)^{-\frac{5}{7}}, \left(ML_1^{\frac{2}{3}}L_2^{\frac{1}{3}}\right)^{-\frac{3}{7}}\right\}\right)$, $T = \Theta\left(\min\left\{\max\left\{\left\lceil\left(\frac{\sigma}{L_2 D^2}\right)^{\frac{2}{5}}\right\rceil, \left\lceil\left(\frac{L_1}{L_2 D}\right)^{\frac{1}{3}}\right\rceil\right\}, \frac{M}{2}\right\}\right)$, $K = \lfloor\frac{M}{T}\rfloor$, *and* $\eta = \Theta\left(\frac{1}{\sqrt{L_1^2+\sigma^2 T/D^2}}\right)$ *in Algorithm 1, then*

$$\mathbb{E}\left[\frac{1}{K}\sum_{k=1}^K \|\nabla F(\bar{\mathbf{w}}^k)\|\right] = \mathcal{O}\left((F(\mathbf{x}_0) - F^*)^{\frac{2}{7}}L_2^{\frac{1}{7}}\frac{\sigma^{\frac{4}{7}}}{M^{\frac{2}{7}}} + L_1^{\frac{2}{7}}L_2^{\frac{1}{7}}\frac{(F(\mathbf{x}_0) - F^*)^{\frac{4}{7}}}{M^{\frac{4}{7}}}\right). \quad (7)$$

*Proof Sketch.* Based on Proposition 2.1, achieving the final convergence rate requires bounding the $K$-shifting regret. As a corollary of Lemma 3.2, we get $\mathbb{E}\left[\text{Reg}_T(\mathbf{u}^1, \ldots, \mathbf{u}^K)\right] \leq \frac{4KD^2}{\eta} + \frac{9L_1^2\eta KD^2}{2} + 6\eta KT\sigma^2$. Combining this regret bound with Proposition 2.1, we obtain

$$\mathbb{E}\left[\frac{1}{K}\sum_{k=1}^K \|\nabla F(\bar{\mathbf{w}}^k)\|\right] \leq \frac{F(\mathbf{x}_0) - F^*}{DKT} + \frac{4D}{T\eta} + \frac{9L_1^2\eta D}{2T} + \frac{6\eta}{D}\sigma^2 + \frac{L_2}{48}D^2 + \frac{L_2}{2}T^2 D^2 + \frac{\sigma}{\sqrt{T}}.$$

Finally, by selecting the parameters $\eta$, $D$, and $T$ according to the theorem's prescription, we arrive at the stated convergence rate. The definitions of $D$ and $T$ involve a max and min, respectively, to interpolate between the deterministic and stochastic regimes. This allows the final bound to adapt to the variance $\sigma$ of the stochastic oracle, interpolating between both settings. $\square$

We show that after $M$ iterations, the gradient norm at the best iterate satisfies $\min_{1\leq k\leq K}\mathbb{E}\left[\|\nabla F(\bar{\mathbf{w}}^k)\|\right] = \mathcal{O}\left(\frac{\sigma^{4/7}}{M^{2/7}} + \frac{1}{M^{4/7}}\right)$, which yields a total gradient complexity of $\mathcal{O}(\sigma^2\varepsilon^{-3.5} + \varepsilon^{-1.75})$. This matches the lower bound $\Omega(\sigma^2\varepsilon^{-3.5})$ in the stochastic regime Arjevani et al. (2023) and recovers the best known rate $\mathcal{O}(\varepsilon^{-1.75})$ in the deterministic setting, smoothly interpolating between these two regimes.

*Remark* 3.1. While our paper focuses on the smooth setting, Algorithm 1 also recovers the optimal convergence rate in the nonsmooth setting with properly chosen parameters. In this case, we assume that $\mathbb{E}\left[\|\nabla f(\mathbf{x};\xi)\|^2\right] \leq G^2$ for all $\mathbf{x} \in \mathbb{R}^d$ and follow the stationary notation of Cutkosky et al. (2023). Under this assumption, Algorithm 1 achieves a complexity of $\mathcal{O}(\epsilon^{-3}\delta^{-1})$ for finding $(\delta, \epsilon)$-stationary points. This guarantee is obtained by adapting our analysis to the nonsmooth regime. Specifically, we replace Proposition 2.1 with Theorem 8 of Cutkosky et al. (2023), which serves as its nonsmooth analogue, and in the regret bound of Lemma 3.1 we discard the negative term and bound $\|\mathbf{g}_n - \mathbf{h}_n\|^2$ directly using the bounded second-moment assumption. The remaining steps follow the same structure as the analysis in Cutkosky et al. (2023).

## 4   ADAPTIVE STEP SIZE SCHEME

In the previous section, we analyzed Algorithm 1 with a constant step size $\eta$ and established its convergence rate. However, as shown in Theorem 3.3, the choice of $\eta$ depends on the global gradient Lipschitz constant and the episode length $T$, which can be overly conservative and leads to slow convergence. To address this, it is often desirable to select the step size adaptively to exploit the local Lipschitz continuity of the objective function. In this section, we extend Algorithm 1 to incorporate an adaptive step size scheme that automatically adjusts the step size based on past gradient information.

Inspired by the adaptive step size analyses developed in Kavis et al. (2019); Antonakopoulos et al. (2022); Levy et al. (2018), we define the learning rate as

$$\eta_n = \frac{\gamma D}{\sqrt{\alpha + \sum_{i=1}^{n \bmod T} \|\mathbf{g}_i^k - \mathbf{h}_i^k\|^2}}, \tag{8}$$

where $\gamma > 0$ controls the initial step size and $\alpha > 0$ is a small constant added for numerical stability. Here, $\mathbf{g}_i^k = \mathbf{g}_{(k-1)T+i}$ and $\mathbf{h}_i^k = \mathbf{h}_{(k-1)T+i}$, with $k$ indicating the episode and $i$ the iteration index within that episode. The sum $\sum_{i=1}^{n \bmod T} \|\mathbf{g}_i^k - \mathbf{h}_i^k\|^2$ is reset at the beginning of each episode, which facilitates the theoretical analysis of regret in the Online Learning Problem 1.

The key challenge in analyzing the adaptive step size scheme in (8) is that the step size $\eta_n$ does not admit a direct upper bound, making it difficult to leverage the negative term in the regret analysis. Specifically, similar to Lemma 3.1, we can bound the regret for the first episode as (the subsequent episodes follow by similar arguments) $\sum_{n=1}^{T} \langle \mathbf{g}_n, \boldsymbol{\Delta}_n - \mathbf{u}^1 \rangle \leq \frac{3D^2}{\eta_T} + \sum_{n=1}^{T} \left( \frac{3\eta_n}{2} \|\mathbf{g}_n - \mathbf{h}_n\|^2 - \frac{1}{4\eta_n} \|\boldsymbol{\Delta}_n - \boldsymbol{\Delta}_{n-1}\|^2 \right)$. In the constant step size setting, as shown in the proof of Lemma 3.2, the key to applying the telescoping argument is the ability to choose $\eta \leq \frac{1}{\sqrt{3}L_1}$, ensuring that the negative term $-\frac{1}{4\eta} \|\boldsymbol{\Delta}_n - \boldsymbol{\Delta}_{n-1}\|^2$ balances the deterministic approximation error $\frac{3\eta}{2} \|\nabla F(\mathbf{w}_n) - \nabla F(\mathbf{z}_{n-1})\|^2$. However, in the adaptive setting, we have no guarantee that $\eta_n$ is bounded by $\frac{1}{\sqrt{3}L_1}$, making the telescoping argument not directly applicable.

To overcome this difficulty, we adopt a thresholding strategy inspired by prior work in Kavis et al. (2019); Antonakopoulos et al. (2022); Levy et al. (2018). At a high level, we partition the iterations within each episode into two categories. For iterations where $\eta_n$ is below a certain threshold, we show that the negative term dominates the positive one. Conversely, when $\eta_n$ is large, the definition of (8) implies that the denominator $\sum_{i=1}^{n \bmod T} \|\mathbf{g}_i^k - \mathbf{h}_i^k\|^2$ is small, which allows us to control the cumulative contribution of the positive terms. We refer the reader to the appendix for further details.

Under the adaptive step size $\eta_n$, we establish the following lemma, which bounds the regret incurred during a single episode of the $K$-shifting regret. Without loss of generality, we present the result for the first episode; the analysis extends straightforwardly to any subsequent episode.

**Lemma 4.1.** *Consider executing the update described in* (5), *using* $\mathbf{h}_n$ *as defined in* (6) *(or its stochastic counterpart in the stochastic setting), and the adaptive step size defined in* (8). *Then we can bound the first episode of the $K$-shifting regret as*

$$\sum_{n=1}^{T} \mathbb{E}\left[ \langle \mathbf{g}_n, \boldsymbol{\Delta}_n - \mathbf{u}^1 \rangle \right] \leq 8 \left( \frac{3}{\gamma} + \gamma \right) D\sqrt{T}\sigma + 16 \left( \frac{3}{\gamma} + \gamma \right)^{\frac{3}{2}} \hat{L}_1 \gamma^{\frac{1}{2}} D^2,$$

*where* $\hat{L}_1 = \max_{1 \leq n \leq T} \frac{\|\mathbf{g}_n - \mathbf{h}_n\|}{\|\mathbf{w}_n - \mathbf{z}_{n-1}\|}$.

Lemma 4.1 plays the same role as Lemma 3.2 in the constant step size setting. Notably, instead of relying on the global gradient Lipschitz constant, the bound depends on $\hat{L}_1$, which can be interpreted as a local Lipschitz constant and may be significantly smaller in practice. We are now ready to present the final convergence guarantee for this adaptive variant of the "Online Doubly Optimistic Gradient" method.

**Theorem 4.2.** *Suppose Assumptions 2.3, 2.1, and 2.2 hold. If we run Algorithm 1 with $\mathbf{h}_n$ as described in* (6) *(or its stochastic counterpart in the stochastic setting), the values for $D$, $T$, and $K$ are selected as the ones in Theorem 3.3, and $\eta_n$ as described in* (8), *replacing $L_1$ with $\hat{L}_1^*$, where*

$\hat{L}_1^* = \max_{1 \leq n \leq M} \frac{\|\mathbf{g}_n - \mathbf{h}_n\|}{\|\mathbf{w}_n - \mathbf{z}_{n-1}\|}$, *then the following holds*

$$\mathbb{E}\left[\frac{1}{K}\sum_{k=1}^{K}\|\nabla F(\bar{\mathbf{w}}^k)\|\right] = \mathcal{O}\left((F(\mathbf{x}_0) - F^*)^{\frac{2}{7}} L_2^{\frac{1}{7}} \frac{\sigma^{\frac{4}{7}}}{M^{\frac{2}{7}}} + (\hat{L}_1^*)^{\frac{2}{7}} L_2^{\frac{1}{7}} \frac{(F(\mathbf{x}_0) - F^*)^{\frac{4}{7}}}{M^{\frac{4}{7}}}\right). \quad (9)$$

Theorem 4.2 shows that our adaptive step size scheme maintains the same complexity of $\mathcal{O}(\varepsilon^{-1.75} + \sigma^2 \varepsilon^{-3.5})$ without requiring manual tuning of the learning rate, while offering improved dependence on the gradient Lipschitz constant compared to the constant step size scheme. Moreover, Cutkosky & Mehta (2020) posed an open problem of designing an algorithm that achieves this complexity without knowledge of the variance $\sigma^2$. Our result represents a first step towards this goal by introducing an adaptive step size. A promising future direction is to further adapt the choices of $D$ and $T$, ultimately aiming to make the algorithm fully parameter-free.

## 5 CONCLUSION

In this paper, we build on the online-to-nonconvex conversion framework of Cutkosky et al. (2023) and propose a simple algorithm for solving smooth nonconvex optimization problems, which only requires two (stochastic) gradient queries per iteration. Our key contribution is a doubly optimistic hint function for the online optimistic gradient method, thus eliminating the fixed point iteration subroutine from (Cutkosky et al., 2023). Assuming that both the gradient and the Hessian of the objective are Lipschitz continuous and that the stochastic gradient has bounded variance $\sigma^2$, our algorithm achieves a complexity of $\mathcal{O}(\varepsilon^{-1.75} + \sigma^2 \varepsilon^{-3.5})$. This result smoothly interpolates between the best-known deterministic rate and the worst-case optimal stochastic rate. In addition, we develop an adaptive step size strategy for our algorithm, marking a first step toward a potential future direction: the design of fully adaptive, parameter-free algorithms that match this performance without relying on prior knowledge of problem-specific parameters.

### ACKNOWLEDGMENTS

This work was supported in part by the NSF CAREER Award CCF-2338846, the NSF AI Institute for Foundations of Machine Learning (IFML), and the NSF AI Institute for Future Edge Networks and Distributed Intelligence (AI-EDGE).

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

APPENDIX

## A  MORE DETAILS ON ONLINE-TO-NONCONVEX CONVERSION AND RELATED PROOFS

In this section, we elaborate on the details of the online-to-nonconvex conversion under the assumption that we have access to a deterministic oracle, with the goal of building intuition. In the next subsection, these results will be extended to the case where only a stochastic oracle is available.

As introduced in Section 2.1, to eliminate the randomness from the analysis, we define the gradient estimate $\mathbf{g}_n = \nabla F(\mathbf{w}_n)$ where $\mathbf{w}_n = \frac{1}{2}(\mathbf{x}_{n-1} + \mathbf{x}_n)$. This modification introduces an error in approximating $F(\mathbf{x}_n) - F(\mathbf{x}_{n-1})$ by $\mathbf{g}_n^\top \mathbf{\Delta}_n$. However, as established in the next lemma, under Assumption 2.2 and for sufficiently small $D$, this error becomes negligible. The proof is available in (Jiang et al., 2025, Appendix A.1).

**Lemma A.1.** *Consider* $\mathbf{x}_n = \mathbf{x}_{n-1} + \mathbf{\Delta}_n$ *where* $\|\mathbf{\Delta}_n\| \leq D$. *Further, define* $\mathbf{g}_n = \nabla F(\mathbf{w}_n)$ *where* $\mathbf{w}_n = \frac{1}{2}(\mathbf{x}_{n-1} + \mathbf{x}_n)$. *If Assumption 2.2 holds, then* $F(\mathbf{x}_{n-1}) - F(\mathbf{x}_n) \geq -\mathbf{g}_n^\top \mathbf{\Delta}_n - \frac{L_2 D^3}{48}$.

Lemma A.1 provides a formal connection between $\mathbf{g}_n$, $\mathbf{\Delta}_n$, and the function value decrease. While the choice $-\mathbf{g}_n$ yields the steepest local descent, as discussed in Section 2.1, the key challenge is that $\mathbf{g}_n$ cannot be computed prior to selecting $\mathbf{\Delta}_n$, since it depends on $\mathbf{x}_n$, which itself requires $\mathbf{\Delta}_n$. This observation motivates an alternative interpretation: the process of choosing $\mathbf{\Delta}_n$ can be cast as an instance of *online learning*, where each $\mathbf{\Delta}_n$ serves as a decision variable, and the loss incurred at step $n$ is given by the linear form $\mathbf{g}_n^\top \mathbf{\Delta}_n$.

To rigorously relate the online learning perspective of choosing $\mathbf{\Delta}_n$ to the objective of identifying a stationary point, we appeal to Lemma A.1. This result establishes that for any fixed comparator $\mathbf{u}$, after $T$ iterations satisfies the following bound:

$$F(\mathbf{x}_T) - F(\mathbf{x}_0) \leq \sum_{n=1}^{T} \mathbf{g}_n^\top (\mathbf{\Delta}_n - \mathbf{u}) + \sum_{n=1}^{T} \mathbf{g}_n^\top \mathbf{u} + \frac{T L_2 D^3}{48}.$$

Setting the arbitrary vector as $\mathbf{u} = -D(\sum_{n=1}^{T} \mathbf{g}_n)/\|\sum_{n=1}^{T} \mathbf{g}_n\|$, it can be shown:

$$\left\|\frac{1}{T}\sum_{n=1}^{T} \mathbf{g}_n\right\| \leq \frac{F(\mathbf{x}_0) - F(\mathbf{x}_T)}{DT} + \frac{1}{DT}\sum_{n=1}^{T} \mathbf{g}_n^\top (\mathbf{\Delta}_n - \mathbf{u}) + \frac{L_2 D^2}{48}. \tag{10}$$

While the connection between the average gradient and the cumulative regret over $T$ iterations has been established, our primary objective remains the identification of an $\varepsilon$-first-order stationary point. To bridge this gap, we now present a formal relationship linking the average of the gradients to the gradient evaluated at the mean iterate. The following lemma serves a similar purpose to (Cutkosky et al., 2023, Proposition 15).

**Lemma A.2.** *If* $\mathbf{g}_n = \nabla F(\mathbf{w}_n)$ *and Assumption 2.2 holds, then* $\|\nabla F(\bar{\mathbf{w}})\| \leq \|\frac{1}{t}\sum_{n=1}^{t} \mathbf{g}_n\| + \frac{L_2}{2} t^2 D^2$, *where* $\bar{\mathbf{w}} = \frac{1}{t}\sum_{n=1}^{t} \mathbf{w}_n$.

The proof of this lemma is provided in (Jiang et al., 2025, Appendix A.2). By combining this result with the expression in (10), we obtain a connection between the norm of the gradient at the averaged iterate and the regret bound appearing on the right-hand side. It is important to note that the error incurred when approximating the average of the gradients over $t$ iterations by the gradient evaluated at the average iterate grows quadratically with the window length $t$. To control this approximation error, we reset the averaging process every $T$ iterations. At the same time, we must account for the fact that certain terms in the upper bound scale inversely with $T$, requiring a careful choice of $T$ to achieve the tightest possible convergence guarantees.

### A.1  PROOF OF PROPOSITION 2.1

Note that when we have access to a stochastic oracle, the gradient becomes $\mathbf{g}_n = \nabla f(\mathbf{w}_n, \xi_n)$. Consequently, Lemma A.1 takes the following form:

$$F(\mathbf{x}_{n-1}) - F(\mathbf{x}_n) \geq -\nabla F(\mathbf{w}_n)^\top \mathbf{\Delta}_n - \frac{L_2 D^3}{48}.$$

This expression can be further related to $\mathbf{g}_n$ as follows:

$$F(\mathbf{x}_{n-1}) - F(\mathbf{x}_n) \geq -\langle \nabla F(\mathbf{w}_n) - \mathbf{g}_n, \boldsymbol{\Delta}_n \rangle - \mathbf{g}_n^\top \boldsymbol{\Delta}_n - \frac{L_2 D^3}{48}.$$

Taking expectations and using the fact that $\mathbf{g}_n$ is an unbiased estimator of $\nabla F(\mathbf{w}_n)$, we get:

$$\mathbb{E}[F(\mathbf{x}_{n-1})] - \mathbb{E}[F(\mathbf{x}_n)] \geq -\mathbb{E}[\mathbf{g}_n^\top \boldsymbol{\Delta}_n] - \frac{L_2 D^3}{48}.$$

By telescoping over $T$ iterations and adding and subtracting $\sum_{n=1}^T \mathbf{g}_n^\top \mathbf{u}$:

$$\mathbb{E}[F(\mathbf{x}_T)] - F(\mathbf{x}_0) \leq \mathbb{E}\left[\sum_{n=1}^T \langle \mathbf{g}_n, \boldsymbol{\Delta}_n - \mathbf{u} \rangle\right] + \mathbb{E}\left[\sum_{n=1}^T \mathbf{g}_n^\top \mathbf{u}\right] + \frac{T L_2 D^3}{48}.$$

Choosing the comparator vector as $\mathbf{u} = -D(\sum_{n=1}^T \mathbf{g}_n)/\|\sum_{n=1}^T \mathbf{g}_n\|$, we obtain the following bound

$$\mathbb{E}\left[\left\|\frac{1}{T}\sum_{n=1}^T \mathbf{g}_n\right\|\right] \leq \frac{F(\mathbf{x}_0) - \mathbb{E}[F(\mathbf{x}_T)]}{DT} + \frac{1}{DT}\mathbb{E}\left[\sum_{n=1}^T \mathbf{g}_n^\top(\boldsymbol{\Delta}_n - \mathbf{u})\right] + \frac{L_2 D^2}{48}. \quad (11)$$

Now consider the $k$-th episode ($k = 1, 2, \ldots, K$) from $n = (k-1)T + 1$ to $n = kT$. By applying the inequality in (11), we obtain

$$\mathbb{E}\left[\left\|\frac{1}{T}\sum_{n=(k-1)T+1}^{kT} \mathbf{g}_n\right\|\right] \leq \frac{\mathbb{E}[F(\mathbf{x}_{(k-1)T})] - \mathbb{E}[F(\mathbf{x}_{kT})]}{DT} + \frac{1}{DT}\mathbb{E}\left[\sum_{n=(k-1)T+1}^{kT} \mathbf{g}_n^\top(\boldsymbol{\Delta}_n - \mathbf{u}^k)\right]$$
$$+ \frac{L_2 D^2}{48}. \quad (12)$$

Moreover, recall that $\bar{\mathbf{w}}^k = \frac{1}{T}\sum_{n=(k-1)T+1}^{kT} \mathbf{w}_n$ and it follows from Lemma A.2 that $\|\nabla F(\bar{\mathbf{w}}^k)\| \leq \left\|\frac{1}{T}\sum_{n=(k-1)T+1}^{kT} \nabla F(\mathbf{w}_n)\right\| + \frac{L_2}{2}T^2 D^2$. Recall that Lemma A.2 was derived under the deterministic setting where $\mathbf{g}_n = \nabla F(\mathbf{w}_n)$, but now $\mathbf{g}_n = \nabla f(\mathbf{w}_n, \xi)$. However, we can state:

$$\|\nabla F(\bar{\mathbf{w}}^k)\| \leq \left\|\frac{1}{T}\sum_{n=(k-1)T+1}^{kT} \nabla F(\mathbf{w}_n)\right\| + \frac{L_2}{2}T^2 D^2$$
$$= \left\|\frac{1}{T}\sum_{n=(k-1)T+1}^{kT} (\nabla F(\mathbf{w}_n) - \mathbf{g}_n + \mathbf{g}_n)\right\| + \frac{L_2}{2}T^2 D^2$$
$$\leq \left\|\frac{1}{T}\sum_{n=(k-1)T+1}^{kT} (\nabla F(\mathbf{w}_n) - \mathbf{g}_n)\right\| + \left\|\frac{1}{T}\sum_{n=(k-1)T+1}^{kT} \mathbf{g}_n\right\| + \frac{L_2}{2}T^2 D^2.$$

The final inequality follows from the triangle inequality. In expectation, we can state

$$\mathbb{E}\left[\|\nabla F(\bar{\mathbf{w}}^k)\|\right] \leq \frac{\sigma}{\sqrt{T}} + \mathbb{E}\left[\left\|\frac{1}{T}\sum_{n=(k-1)T+1}^{kT} \mathbf{g}_n\right\|\right] + \frac{L_2}{2}T^2 D^2. \quad (13)$$

This bound follows from applying Assumption 2.3, which allows us to establish $\mathbb{E}\left[\left\|\frac{1}{T}\sum_{n=(k-1)T+1}^{kT} \nabla F(\mathbf{w}_n) - \mathbf{g}_n\right\|\right] \leq \frac{\sigma}{\sqrt{T}}$.

Using (12) together with (13) leads to

$$\mathbb{E}\left[\|\nabla F(\bar{\mathbf{w}}^k)\|\right] \leq \frac{\mathbb{E}\left[F(\mathbf{x}_{(k-1)T})\right] - \mathbb{E}\left[F(\mathbf{x}_{kT})\right]}{DT} + \frac{1}{DT}\mathbb{E}\left[\sum_{n=(k-1)T+1}^{kT} \mathbf{g}_n^\top(\boldsymbol{\Delta}_n - \mathbf{u}^k)\right]$$
$$+ \frac{L_2 D^2}{48} + \frac{L_2}{2}T^2 D^2 + \frac{\sigma}{\sqrt{T}}.$$

Summing over $k = 1$ to $K$ and dividing by $K$, we obtain

$$\frac{1}{K}\sum_{k=1}^{K}\mathbb{E}\left[\|\nabla F(\bar{\mathbf{w}}^k)\|\right] \leq \frac{F(\mathbf{x}_0) - \mathbb{E}\left[F(\mathbf{x}_M)\right]}{KDT} + \frac{1}{KDT}\mathbb{E}\left[\sum_{k=1}^{K}\sum_{n=(k-1)T+1}^{kT}\mathbf{g}_n^\top(\boldsymbol{\Delta}_n - \mathbf{u}^k)\right]$$
$$+ \frac{L_2 D^2}{48} + \frac{L_2}{2}T^2 D^2 + \frac{\sigma}{\sqrt{T}}.$$

Finally, noting that $\mathbb{E}\left[F(\mathbf{x}_M)\right] \geq F^*$ completes the proof.

## B    PROOFS FOR ONLINE DOUBLY OPTIMISTIC GRADIENT WITH CONSTANT STEP SIZE

In this section, we provide the detailed proofs corresponding to Section 3, where the *Online Doubly Optimistic Gradient* method with a constant step size was introduced, along with its complexity analysis. While some of the proofs closely follow arguments from Jiang et al. (2025), we include them here for completeness.

### B.1    PROOF OF LEMMA 3.1

To prepare for the proof of Lemma 3.1, we first prove an auxiliary result.

**Lemma B.1.** *Consider the update rule in* (5). *Then for any* $\mathbf{u}$ *such that* $\|\mathbf{u}\| \leq D$, *it holds that*

$$\langle \mathbf{g}_{n+1}, \boldsymbol{\Delta}_{n+1} - \mathbf{u}\rangle \leq \frac{\|\boldsymbol{\Delta}_n - \mathbf{u}\|^2}{2\eta} - \frac{\|\boldsymbol{\Delta}_{n+1} - \mathbf{u}\|^2}{2\eta} + \langle \mathbf{g}_{n+1} - \mathbf{h}_{n+1}, \boldsymbol{\Delta}_{n+1} - \mathbf{u}\rangle$$
$$- \langle \mathbf{g}_n - \mathbf{h}_n, \boldsymbol{\Delta}_n - \mathbf{u}\rangle + \eta\|\mathbf{g}_n - \mathbf{h}_n\|^2 - \frac{1}{4\eta}\|\boldsymbol{\Delta}_{n+1} - \boldsymbol{\Delta}_n\|^2.$$

*Proof.* By using the optimality condition, for any $\mathbf{u} \in \{\boldsymbol{\Delta} \in \mathbb{R}^d : \|\boldsymbol{\Delta}\| \leq D\}$, we have

$$\langle \boldsymbol{\Delta}_{n+1} - \boldsymbol{\Delta}_n + \eta\mathbf{h}_{n+1} + \eta(\mathbf{g}_n - \mathbf{h}_n), \mathbf{u} - \boldsymbol{\Delta}_{n+1}\rangle \geq 0.$$

Rearranging terms, we obtain

$$\langle \mathbf{g}_{n+1}, \boldsymbol{\Delta}_{n+1} - \mathbf{u}\rangle \leq \langle \mathbf{g}_{n+1} - \mathbf{h}_{n+1}, \boldsymbol{\Delta}_{n+1} - \mathbf{u}\rangle - \langle \mathbf{g}_n - \mathbf{h}_n, \boldsymbol{\Delta}_{n+1} - \mathbf{u}\rangle$$
$$+ \frac{1}{2\eta}\|\boldsymbol{\Delta}_n - \mathbf{u}\|^2 - \frac{1}{2\eta}\|\boldsymbol{\Delta}_{n+1} - \boldsymbol{\Delta}_n\|^2 - \frac{1}{2\eta}\|\boldsymbol{\Delta}_{n+1} - \mathbf{u}\|^2,$$

where we have used the three-point equality $\langle \boldsymbol{\Delta}_{n+1} - \boldsymbol{\Delta}_n, \mathbf{u} - \boldsymbol{\Delta}_{n+1}\rangle = \frac{1}{2}\|\boldsymbol{\Delta}_n - \mathbf{u}\|^2 - \frac{1}{2}\|\boldsymbol{\Delta}_{n+1} - \boldsymbol{\Delta}_n\|^2 - \frac{1}{2}\|\boldsymbol{\Delta}_{n+1} - \mathbf{u}\|^2$. Moreover, we have:

$$-\langle \mathbf{g}_n - \mathbf{h}_n, \boldsymbol{\Delta}_{n+1} - \mathbf{u}\rangle = -\langle \mathbf{g}_n - \mathbf{h}_n, \boldsymbol{\Delta}_n - \mathbf{u}\rangle + \langle \mathbf{g}_n - \mathbf{h}_n, \boldsymbol{\Delta}_n - \boldsymbol{\Delta}_{n+1}\rangle.$$

We can further bound the second term as follows:

$$\langle \mathbf{g}_n - \mathbf{h}_n, \boldsymbol{\Delta}_n - \boldsymbol{\Delta}_{n+1}\rangle \leq \|\mathbf{g}_n - \mathbf{h}_n\|\|\boldsymbol{\Delta}_n - \boldsymbol{\Delta}_{n+1}\| \leq \eta\|\mathbf{g}_n - \mathbf{h}_n\|^2 + \frac{1}{4\eta}\|\boldsymbol{\Delta}_{n+1} - \boldsymbol{\Delta}_n\|^2,$$

where the final inequality follows from the weighted Young's inequality. Putting all of the above inequalities together yields the desired bound. □

Using Lemma B.1, we can bound the regret for each episode in the following lemma.

**Lemma B.2.** *Consider the optimistic update in* (5). *Then for* $k = 1$ *and any* $\mathbf{u}^1$ *such that* $\|\mathbf{u}^1\| \leq D$, *we have*

$$\sum_{n=1}^{T}\langle \mathbf{g}_n, \boldsymbol{\Delta}_n - \mathbf{u}^1\rangle \leq \frac{2D^2}{\eta} + \sum_{n=1}^{T}\eta\|\mathbf{g}_n - \mathbf{h}_n\|^2 - \sum_{n=2}^{T}\frac{1}{4\eta}\|\boldsymbol{\Delta}_n - \boldsymbol{\Delta}_{n-1}\|^2. \qquad (14)$$

*For $k \geq 2$ and any $\mathbf{u}^k$ such that $\|\mathbf{u}^k\| \leq D$, we have*

$$\sum_{n=(k-1)T+1}^{kT} \langle \mathbf{g}_n, \boldsymbol{\Delta}_n - \mathbf{u}^k \rangle \leq \frac{4D^2}{\eta} + \frac{\eta}{2} \left\| \mathbf{g}_{(k-1)T} - \mathbf{h}_{(k-1)T} \right\|^2 + \sum_{n=(k-1)T}^{kT} \eta \|\mathbf{g}_n - \mathbf{h}_n\|^2$$
$$- \sum_{n=(k-1)T+1}^{kT} \frac{1}{4\eta} \|\boldsymbol{\Delta}_n - \boldsymbol{\Delta}_{n-1}\|^2. \tag{15}$$

*Proof.* To prove (14), we set $\mathbf{u} = \mathbf{u}^1$ and sum the inequality in Lemma B.1 from $n = 1$ to $n = T - 1$:

$$\sum_{n=2}^{T} \langle \mathbf{g}_n, \boldsymbol{\Delta}_n - \mathbf{u}^1 \rangle \leq \frac{1}{2\eta} \|\boldsymbol{\Delta}_1 - \mathbf{u}^1\|^2 - \frac{1}{2\eta} \|\boldsymbol{\Delta}_T - \mathbf{u}^1\|^2 + \langle \mathbf{g}_T - \mathbf{h}_T, \boldsymbol{\Delta}_T - \mathbf{u}^1 \rangle$$
$$- \langle \mathbf{g}_1 - \mathbf{h}_1, \boldsymbol{\Delta}_1 - \mathbf{u}^1 \rangle + \sum_{n=1}^{T-1} \left( \eta \|\mathbf{g}_n - \mathbf{h}_n\|^2 - \frac{1}{4\eta} \|\boldsymbol{\Delta}_{n+1} - \boldsymbol{\Delta}_n\|^2 \right).$$

Moreover, recall that $\boldsymbol{\Delta}_1 = \arg\min_{\|\boldsymbol{\Delta}\| \leq D} \langle \mathbf{h}_1, \boldsymbol{\Delta} \rangle$. Thus, this implies that $\langle \mathbf{h}_1, \boldsymbol{\Delta}_1 - \mathbf{u}^1 \rangle \leq 0$.

$$\langle \mathbf{g}_T - \mathbf{h}_T, \boldsymbol{\Delta}_T - \mathbf{u}^1 \rangle \leq \|\mathbf{g}_T - \mathbf{h}_T\| \|\boldsymbol{\Delta}_T - \mathbf{u}^1\| \leq \frac{\eta}{2} \|\mathbf{g}_T - \mathbf{h}_T\|^2 + \frac{1}{2\eta} \|\boldsymbol{\Delta}_T - \mathbf{u}^1\|^2.$$

Combining all the inequalities, we obtain that

$$\sum_{n=1}^{T} \langle \mathbf{g}_n, \boldsymbol{\Delta}_n - \mathbf{u}^1 \rangle \leq \frac{1}{2\eta} \|\boldsymbol{\Delta}_1 - \mathbf{u}^1\|^2 + \frac{\eta}{2} \|\mathbf{g}_T - \mathbf{h}_T\|^2 + \sum_{n=1}^{T-1} \left( \eta \|\mathbf{g}_n - \mathbf{h}_n\|^2 - \frac{1}{4\eta} \|\boldsymbol{\Delta}_{n+1} - \boldsymbol{\Delta}_n\|^2 \right).$$

Using $\frac{\eta}{2} \|\mathbf{g}_T - \mathbf{h}_T\|^2 \leq \eta \|\mathbf{g}_T - \mathbf{h}_T\|^2$ and $\sum_{n=1}^{T-1} \frac{1}{4\eta} \|\boldsymbol{\Delta}_{n+1} - \boldsymbol{\Delta}_n\|^2 = \sum_{n=2}^{T} \frac{1}{4\eta} \|\boldsymbol{\Delta}_n - \boldsymbol{\Delta}_{n-1}\|^2$ lead to:

$$\sum_{n=1}^{T} \langle \mathbf{g}_n, \boldsymbol{\Delta}_n - \mathbf{u}^1 \rangle \leq \frac{1}{2\eta} \|\boldsymbol{\Delta}_1 - \mathbf{u}^1\|^2 + \sum_{n=1}^{T} \eta \|\mathbf{g}_n - \mathbf{h}_n\|^2 - \sum_{n=2}^{T} \frac{1}{4\eta} \|\boldsymbol{\Delta}_n - \boldsymbol{\Delta}_{n-1}\|^2.$$

Finally, since we have $\|\boldsymbol{\Delta}_1\| = D$ and $\|\mathbf{u}^1\| \leq D$, this leads to (14).

Now we move to (15). To simplify the notation, we let $\boldsymbol{\Delta}_t^k := \boldsymbol{\Delta}_{(k-1)T+t}$, $\mathbf{g}_t^k := \mathbf{g}_{(k-1)T+t}$ and $\mathbf{h}_t^k = \mathbf{h}_{(k-1)T+t}$. Then by setting $\mathbf{u} = \mathbf{u}^k$ and summing the inequality in Lemma B.1 from $n = (k-1)T$ to $n = kT - 1$, we obtain:

$$\sum_{t=1}^{T} \langle \mathbf{g}_t^k, \boldsymbol{\Delta}_t^k - \mathbf{u}^k \rangle \leq \frac{\|\boldsymbol{\Delta}_0^k - \mathbf{u}^k\|^2}{2\eta} - \frac{\|\boldsymbol{\Delta}_T^k - \mathbf{u}^k\|^2}{2\eta} + \langle \mathbf{g}_T^k - \mathbf{h}_T^k, \boldsymbol{\Delta}_T^k - \mathbf{u}^k \rangle$$
$$- \langle \mathbf{g}_T^{k-1} - \mathbf{h}_T^{k-1}, \boldsymbol{\Delta}_T^{k-1} - \mathbf{u}^k \rangle + \sum_{n=(k-1)T}^{kT-1} \left( \eta \|\mathbf{g}_n - \mathbf{h}_n\|^2 - \frac{1}{4\eta} \|\boldsymbol{\Delta}_{n+1} - \boldsymbol{\Delta}_n\|^2 \right).$$

Following a similar argument as in the proof of (14), we have the following inequalities:

- $\langle \mathbf{g}_T^k - \mathbf{h}_T^k, \boldsymbol{\Delta}_T^k - \mathbf{u}^k \rangle \leq \frac{\eta}{2} \left\| \mathbf{g}_T^k - \mathbf{h}_T^k \right\|^2 + \frac{1}{2\eta} \|\boldsymbol{\Delta}_T^k - \mathbf{u}^k\|^2$.

- $\langle \mathbf{g}_T^{k-1} - \mathbf{h}_T^{k-1}, \boldsymbol{\Delta}_T^{k-1} - \mathbf{u}^k \rangle \leq \frac{\eta}{2} \left\| \mathbf{g}_T^{k-1} - \mathbf{h}_T^{k-1} \right\|^2 + \frac{1}{2\eta} \|\boldsymbol{\Delta}_T^{k-1} - \mathbf{u}^k\|^2$.

- $\sum_{n=(k-1)T}^{kT-1} \frac{1}{4\eta} \|\boldsymbol{\Delta}_{n+1} - \boldsymbol{\Delta}_n\|^2 = \sum_{n=(k-1)T+1}^{kT} \frac{1}{4\eta} \|\boldsymbol{\Delta}_n - \boldsymbol{\Delta}_{n-1}\|^2$.

Thus, combining all the inequalities above, we obtain:

$$\sum_{t=1}^{T}\langle \mathbf{g}_t^k, \boldsymbol{\Delta}_t^k - \mathbf{u}^k\rangle \leq \frac{1}{2\eta}\|\boldsymbol{\Delta}_0^k - \mathbf{u}^k\|^2 + \frac{\eta}{2}\left\|\mathbf{g}_T^k - \mathbf{h}_T^k\right\|^2 + \frac{\eta}{2}\left\|\mathbf{g}_T^{k-1} - \mathbf{h}_T^{k-1}\right\|^2 + \frac{1}{2\eta}\|\boldsymbol{\Delta}_T^{k-1} - \mathbf{u}^k\|^2$$

$$+ \sum_{n=(k-1)T}^{kT-1}\eta\|\mathbf{g}_n - \mathbf{h}_n\|^2 - \sum_{n=(k-1)T+1}^{kT}\frac{1}{4\eta}\|\boldsymbol{\Delta}_n - \boldsymbol{\Delta}_{n-1}\|^2$$

$$\leq \frac{\|\boldsymbol{\Delta}_0^k - \mathbf{u}^k\|^2}{2\eta} + \frac{\eta\left\|\mathbf{g}_T^{k-1} - \mathbf{h}_T^{k-1}\right\|^2}{2} + \frac{\|\boldsymbol{\Delta}_T^{k-1} - \mathbf{u}^k\|^2}{2\eta} + \sum_{n=(k-1)T}^{kT}\eta\|\mathbf{g}_n - \mathbf{h}_n\|^2$$

$$- \sum_{n=(k-1)T+1}^{kT}\frac{1}{4\eta}\|\boldsymbol{\Delta}_n - \boldsymbol{\Delta}_{n-1}\|^2.$$

Finally, since we have $\|\boldsymbol{\Delta}_0^k\| \leq D$, $\|\mathbf{u}\| \leq D$, and $\|\boldsymbol{\Delta}_T^{k-1}\| \leq D$, we get $\frac{\|\boldsymbol{\Delta}_0^k - \mathbf{u}^k\|^2}{2\eta} + \frac{\|\boldsymbol{\Delta}_T^{k-1} - \mathbf{u}^k\|^2}{2\eta} \leq \frac{4D^2}{\eta}$. This completes the proof of (15).

$\square$

Now we are ready to prove Lemma 3.1.

*Proof of Lemma 3.1.* Summing the inequality in (14) and the inequality in (15) for $k = 2, 3, \ldots, K$ in Lemma B.2, we have:

$$\text{Reg}_T(\mathbf{u}^1, \ldots, \mathbf{u}^K) = \sum_{k=1}^{K}\sum_{n=1}^{T}\langle \mathbf{g}_n^k, \boldsymbol{\Delta}_n^k - \mathbf{u}^k\rangle$$

$$\leq \frac{4KD^2}{\eta} + \sum_{k=1}^{K}\frac{\eta}{2}\left\|\mathbf{g}_T^{k-1} - \mathbf{h}_T^{k-1}\right\|^2 + \sum_{n=1}^{KT}\eta\|\mathbf{g}_n - \mathbf{h}_n\|^2 - \sum_{n=2}^{KT}\frac{1}{4\eta}\|\boldsymbol{\Delta}_n - \boldsymbol{\Delta}_{n-1}\|^2$$

$$+ \sum_{k=1}^{K}\eta\|\mathbf{g}_T^{k-1} - \mathbf{h}_T^{k-1}\|^2$$

$$\leq \frac{4KD^2}{\eta} + \sum_{k=1}^{K}\frac{3\eta}{2}\left\|\mathbf{g}_T^{k-1} - \mathbf{h}_T^{k-1}\right\|^2 + \sum_{n=1}^{KT}\eta\|\mathbf{g}_n - \mathbf{h}_n\|^2 - \sum_{n=2}^{KT}\frac{1}{4\eta}\|\boldsymbol{\Delta}_n - \boldsymbol{\Delta}_{n-1}\|^2$$

$$\leq \frac{4KD^2}{\eta} + \frac{5\eta}{2}\sum_{n=1}^{KT}\|\mathbf{g}_n - \mathbf{h}_n\|^2 - \sum_{n=2}^{KT}\frac{1}{4\eta}\|\boldsymbol{\Delta}_n - \boldsymbol{\Delta}_{n-1}\|^2.$$

Where in the last inequality we use the fact that $\sum_{k=1}^{K}\frac{3\eta}{2}\left\|\mathbf{g}_T^{k-1} - \mathbf{h}_T^{k-1}\right\|^2 \leq \sum_{n=1}^{kT}\frac{3\eta}{2}\|\mathbf{g}_n - \mathbf{h}_n\|^2$. This completes the proof. $\square$

## B.2   PROOF OF LEMMA 3.2

To prepare for the proof of Lemma 3.2, we first prove an auxiliary result.

**Lemma B.3.** *Let $\mathbf{h}_n$ be defined as in (6). Then the following bound holds*

$$\mathbb{E}\left[\|\mathbf{g}_n - \mathbf{h}_n\|^2\right] \leq 6\sigma^2 + \frac{3L_1^2}{4}\mathbb{E}\|\boldsymbol{\Delta}_n - \boldsymbol{\Delta}_{n-1}\|^2.$$

*Proof.* To bound $\mathbb{E}\|\mathbf{g}_n - \mathbf{h}_n\|^2$, we begin by recalling the definitions of the respective vectors: $\mathbf{g}_n = \nabla f(\mathbf{w}_n; \xi_n)$ and $\mathbf{h}_n = \nabla f(\mathbf{z}_{n-1}; \xi_{n-1})$. Thus,

$$
\begin{aligned}
\mathbb{E}\left[\|\mathbf{g}_n - \mathbf{h}_n\|^2\right] &= \mathbb{E}\left\|\nabla f(\mathbf{w}_n; \xi_n) - \nabla f(\mathbf{z}_{n-1}; \xi_{n-1})\right\|^2 \\
&= \mathbb{E}\|\nabla f(\mathbf{w}_n; \xi_n) - \nabla f(\mathbf{z}_{n-1}; \xi_{n-1}) \\
&\quad + \nabla F(\mathbf{z}_{n-1}) - \nabla F(\mathbf{z}_{n-1}) \\
&\quad + \nabla F(\mathbf{w}_n) - \nabla F(\mathbf{w}_n)\|^2,
\end{aligned}
$$

where the last equality comes from adding and subtracting its deterministic counterparts. Applying Young's inequality

$$
\begin{aligned}
\mathbb{E}\left[\|\mathbf{g}_n - \mathbf{h}_n\|^2\right] &\leq 3\,\mathbb{E}\left[\|\nabla f(\mathbf{z}_{n-1}; \xi_{n-1}) - \nabla F(\mathbf{z}_{n-1})\|^2\right] \\
&\quad + 3\,\mathbb{E}\left[\|\nabla f(\mathbf{w}_n; \xi_n) - \nabla F(\mathbf{w}_n)\|^2\right] \\
&\quad + 3\,\mathbb{E}\left[\|\nabla F(\mathbf{w}_n) - \nabla F(\mathbf{z}_{n-1})\|^2\right] \\
&\leq 6\sigma^2 + \frac{3L_1^2}{4}\,\mathbb{E}\|\boldsymbol{\Delta}_n - \boldsymbol{\Delta}_{n-1}\|^2.
\end{aligned}
$$

The last inequality follows from Assumption 2.3 that allows us to bound the variance of the first two terms, and the last term comes from leveraging Assumption 2.1, $\left\|\nabla F(\mathbf{x}_{n-1} + \frac{1}{2}\boldsymbol{\Delta}_n) - \nabla F(\mathbf{x}_{n-1} + \frac{1}{2}\boldsymbol{\Delta}_{n-1})\right\|^2 \leq L_1^2 \left\|\frac{1}{2}\boldsymbol{\Delta}_n - \frac{1}{2}\boldsymbol{\Delta}_{n-1}\right\|^2$. $\qquad\square$

To prove Lemma 3.2, we first analyze the case $k \geq 2$ and subsequently show that the bound also holds for $k = 1$. Our analysis focuses on the stochastic setting. However, by Lemma B.2, inequality (15) is valid in both the deterministic and stochastic cases. Therefore, we have:

$$
\begin{aligned}
\sum_{n=(k-1)T+1}^{kT} \langle \mathbf{g}_n, \boldsymbol{\Delta}_n - \mathbf{u}^k \rangle &\leq \frac{4D^2}{\eta} + \frac{\eta}{2}\left\|\mathbf{g}_{(k-1)T} - \mathbf{h}_{(k-1)T}\right\|^2 + \sum_{n=(k-1)T}^{kT} \eta\|\mathbf{g}_n - \mathbf{h}_n\|^2 \\
&\quad - \sum_{n=(k-1)T+1}^{kT} \frac{1}{4\eta}\|\boldsymbol{\Delta}_n - \boldsymbol{\Delta}_{n-1}\|^2.
\end{aligned}
$$

Taking expectations on both sides yields

$$
\begin{aligned}
\sum_{n=(k-1)T+1}^{kT} \mathbb{E}\langle \mathbf{g}_n, \boldsymbol{\Delta}_n - \mathbf{u}^k \rangle &\leq \frac{4D^2}{\eta} + \frac{\eta}{2}\,\mathbb{E}\left\|\mathbf{g}_{(k-1)T} - \mathbf{h}_{(k-1)T}\right\|^2 + \sum_{n=(k-1)T}^{kT} \eta\,\mathbb{E}\|\mathbf{g}_n - \mathbf{h}_n\|^2 \\
&\quad - \sum_{n=(k-1)T+1}^{kT} \frac{1}{4\eta}\,\mathbb{E}\|\boldsymbol{\Delta}_n - \boldsymbol{\Delta}_{n-1}\|^2.
\end{aligned}
\tag{16}
$$

Leveraging Lemma B.3

$$
\begin{aligned}
\sum_{n=(k-1)T+1}^{kT} \mathbb{E}\langle \mathbf{g}_n, \boldsymbol{\Delta}_n - \mathbf{u}^k \rangle &\leq \frac{4D^2}{\eta} + \frac{\eta}{2}\left(6\sigma^2 + \frac{3L_1^2}{4}\,\mathbb{E}\|\boldsymbol{\Delta}_{(k-1)T} - \boldsymbol{\Delta}_{(k-1)T-1}\|^2\right) \\
&\quad + \sum_{n=(k-1)T}^{kT} \eta\left(6\sigma^2 + \frac{3L_1^2}{4}\,\mathbb{E}\|\boldsymbol{\Delta}_n - \boldsymbol{\Delta}_{n-1}\|^2\right) \\
&\quad - \sum_{n=(k-1)T+1}^{kT} \frac{1}{4\eta}\,\mathbb{E}\|\boldsymbol{\Delta}_n - \boldsymbol{\Delta}_{n-1}\|^2.
\end{aligned}
$$

Rearranging the terms leads to

$$\sum_{n=(k-1)T+1}^{kT} \mathbb{E}\langle \mathbf{g}_n, \boldsymbol{\Delta}_n - \mathbf{u}^k \rangle \leq \frac{4D^2}{\eta} + 3\sigma^2\eta + \frac{9L_1^2\eta}{8}\,\mathbb{E}\,\|\boldsymbol{\Delta}_{(k-1)T} - \boldsymbol{\Delta}_{(k-1)T-1}\|^2$$

$$+ \sum_{n=(k-1)T}^{kT} 6\sigma^2\eta + \sum_{n=(k-1)T+1}^{kT} \left(\frac{3L_1^2\eta}{4} - \frac{1}{4\eta}\right)\mathbb{E}\,\|\boldsymbol{\Delta}_n - \boldsymbol{\Delta}_{n-1}\|^2.$$

Note that as long as $\eta \leq \frac{1}{\sqrt{3}L_1}$, it holds that $\sum_{n=(k-1)T+1}^{kT}\left(\frac{3L_1^2\eta}{4} - \frac{1}{4\eta}\right)\mathbb{E}\,\|\boldsymbol{\Delta}_n - \boldsymbol{\Delta}_{n-1}\|^2 \leq 0$.
Finally, since $\|\boldsymbol{\Delta}_n\| \leq D$ for all $n$, we have:

$$\sum_{n=(k-1)T+1}^{kT} \mathbb{E}\langle \mathbf{g}_n, \boldsymbol{\Delta}_n - \mathbf{u}^k \rangle \leq \frac{4D^2}{\eta} + \frac{9L_1^2 D^2\eta}{2} + \left(T + \frac{3}{2}\right)6\sigma^2\eta$$

$$\leq \frac{4D^2}{\eta} + \frac{9L_1^2 D^2\eta}{2} + 18\sigma^2 T\eta, \tag{17}$$

where the last inequality follows from the fact that $T \geq 1$, completing the proof.

For completeness, we analyze the case $k = 1$. As will be shown, the upper bound (17) also holds in this case. From Lemma B.2, inequality (14) gives

$$\sum_{n=1}^{T}\langle \mathbf{g}_n, \boldsymbol{\Delta}_n - \mathbf{u}^1\rangle \leq \frac{2D^2}{\eta} + \sum_{n=1}^{T}\eta\|\mathbf{g}_n - \mathbf{h}_n\|^2 - \sum_{n=2}^{T}\frac{1}{4\eta}\|\boldsymbol{\Delta}_n - \boldsymbol{\Delta}_{n-1}\|^2.$$

Taking expectations on both sides yields

$$\sum_{n=1}^{T}\mathbb{E}\langle \mathbf{g}_n, \boldsymbol{\Delta}_n - \mathbf{u}^1\rangle \leq \frac{2D^2}{\eta} + \sum_{n=1}^{T}\eta\,\mathbb{E}\,\|\mathbf{g}_n - \mathbf{h}_n\|^2 - \sum_{n=2}^{T}\frac{1}{4\eta}\,\mathbb{E}\,\|\boldsymbol{\Delta}_n - \boldsymbol{\Delta}_{n-1}\|^2. \tag{18}$$

Leveraging Lemma B.3

$$\sum_{n=1}^{T}\mathbb{E}\langle \mathbf{g}_n, \boldsymbol{\Delta}_n - \mathbf{u}^1\rangle \leq \frac{2D^2}{\eta} + \sum_{n=1}^{T}\eta\left(6\sigma^2 + \frac{3L_1^2}{4}\,\mathbb{E}\,\|\boldsymbol{\Delta}_n - \boldsymbol{\Delta}_{n-1}\|^2\right) - \sum_{n=2}^{T}\frac{1}{4\eta}\,\mathbb{E}\,\|\boldsymbol{\Delta}_n - \boldsymbol{\Delta}_{n-1}\|^2.$$

Rearranging terms (with the convention $\boldsymbol{\Delta}_0 = 0$) leads to

$$\sum_{n=1}^{T}\mathbb{E}\langle \mathbf{g}_n, \boldsymbol{\Delta}_n - \mathbf{u}^1\rangle \leq \frac{2D^2}{\eta} + \sum_{n=1}^{T}6\sigma^2\eta + \frac{3L_1^2\eta}{4}\,\mathbb{E}\,\|\boldsymbol{\Delta}_1\|^2 + \sum_{n=2}^{T}\left(\frac{3L_1^2\eta}{4} - \frac{1}{4\eta}\right)\mathbb{E}\,\|\boldsymbol{\Delta}_n - \boldsymbol{\Delta}_{n-1}\|^2.$$

Note that as long as $\eta \leq \frac{1}{\sqrt{3}L_1}$, it holds that $\sum_{n=(k-1)T+2}^{kT}\left(\frac{3L_1^2\eta}{4} - \frac{1}{4\eta}\right)\mathbb{E}\,\|\boldsymbol{\Delta}_n - \boldsymbol{\Delta}_{n-1}\|^2 \leq 0$.
Finally, since $\|\boldsymbol{\Delta}_n\| \leq D$ for all $n$, we have:

$$\sum_{n=(k-1)T+1}^{kT} \mathbb{E}\langle \mathbf{g}_n, \boldsymbol{\Delta}_n - \mathbf{u}^k \rangle \leq \frac{2D^2}{\eta} + \frac{3L_1^2 D^2\eta}{4} + 6T\sigma^2\eta$$

$$\leq \frac{4D^2}{\eta} + \frac{9L_1^2 D^2\eta}{2} + 18\sigma^2 T\eta,$$

where the last inequality simply enlarges constants. This confirms that (17) also holds when $k = 1$.

### B.3    Proof of Theorem 3.3

According to Proposition 2.1, attaining the final convergence rate necessitates bounding the $K$-shifting regret. We obtain the following result as a direct consequence of Lemma 3.2

$$\mathbb{E}\left[\mathrm{Reg}_T(\mathbf{u}^1, \ldots, \mathbf{u}^K)\right] \leq \frac{4KD^2}{\eta} + \frac{9L_1^2\eta KD^2}{2} + 18\eta KT\sigma^2.$$

Integrating the aforementioned regret bound with Proposition 2.1, we arrive at

$$\mathbb{E}\left[\frac{1}{K}\sum_{k=1}^{K}\|\nabla F(\bar{\mathbf{w}}^k)\|\right] \le \frac{F(\mathbf{x}_0)-F^*}{DKT} + \frac{4D}{T\eta} + \frac{9L_1^2\eta D}{2T} + \frac{18\eta}{D}\sigma^2 + \frac{L_2}{48}D^2 + \frac{L_2}{2}T^2D^2 + \frac{\sigma}{\sqrt{T}}.$$

Rearranging terms

$$\mathbb{E}\left[\frac{1}{K}\sum_{k=1}^{K}\|\nabla F(\bar{\mathbf{w}}^k)\|\right] \le \frac{F(\mathbf{x}_0)-F^*}{DKT} + \frac{4D}{T\eta} + \left(\frac{9L_1^2D}{2T} + \frac{18}{D}\sigma^2\right)\eta + L_2T^2D^2 + \frac{\sigma}{\sqrt{T}}. \quad (19)$$

From the upper bound in (19), we must select the hyperparameters $\eta, D, K, T$, subject to the constraint $KT \le M$, where $M$ denotes the total iteration budget and $K, T \in \mathbb{N}$. To obtain the tightest possible bound, we will first choose $\eta$, then determine $T$ and $K$, and finally set $D$.

To proceed, we balance the two $\eta$-dependent terms to determine the optimal $\eta$, and scale it appropriately so that the condition $\eta \le \frac{1}{\sqrt{3}L_1}$, required by Lemma 3.2, is satisfied. This yields the choice $\eta = \frac{1}{\sqrt{3L_1^2 + \frac{12T}{D^2}\sigma^2}}$. Substituting this expression into the bound gives

$$\mathbb{E}\left[\frac{1}{K}\sum_{k=1}^{K}\|\nabla F(\bar{\mathbf{w}}^k)\|\right] \le \frac{F(\mathbf{x}_0)-F^*}{DKT} + \frac{11D}{2T}\sqrt{3L_1^2 + \frac{12T}{D^2}\sigma^2} + L_2T^2D^2 + \frac{\sigma}{\sqrt{T}}.$$

Applying the inequality $\sqrt{a+b} \le \sqrt{a} + \sqrt{b}$ for $a, b \ge 0$, and simplifying yields

$$\mathbb{E}\left[\frac{1}{K}\sum_{k=1}^{K}\|\nabla F(\bar{\mathbf{w}}^k)\|\right] \le \frac{F(\mathbf{x}_0)-F^*}{DKT} + \frac{10L_1D}{T} + \frac{20\sigma}{\sqrt{T}} + L_2T^2D^2. \quad (20)$$

Note that the bound involves four terms dependent on $T$. If we set $T = \min\left(\max\left(\left\lceil\left(\frac{20\sigma}{L_2D^2}\right)^{\frac{2}{5}}\right\rceil, \left\lceil\left(\frac{10L_1}{L_2D}\right)^{\frac{1}{3}}\right\rceil\right), \frac{M}{2}\right)$ and $K = \lfloor\frac{M}{T}\rfloor$, we can bound the first three terms of the inequality (20) as follows:

- $KT \ge (\frac{M}{T}-1)T \ge M - T \ge M/2$ which implies $\frac{F(\mathbf{x}_0)-F^*}{DKT} \le 2\frac{F(\mathbf{x}_0)-F^*}{DM}$;

- $\frac{10L_1D}{T} \le \max\left(10^{\frac{2}{3}}L_1^{\frac{2}{3}}L_2^{\frac{1}{3}}D^{\frac{4}{3}}, \frac{20L_1D}{M}\right)$;

- $\frac{20\sigma}{\sqrt{T}} \le \max\left(20^{\frac{4}{5}}L_2^{\frac{1}{5}}D^{\frac{2}{5}}\sigma^{\frac{4}{5}}, \frac{20\sqrt{2}\sigma}{\sqrt{M}}\right)$;

The max terms arise from the fact that the second and third terms in (20) are inversely proportional to $T$. Since $T$ is clipped by $M/2$, these terms become inversely proportional to $\min\left(D^{-\frac{1}{3}}, \frac{M}{2}\right)$ and $\sqrt{\min\left((\frac{\sigma}{D^2})^{\frac{2}{5}}, \frac{M}{2}\right)}$, respectively.

We upper bound $L_2T^2D^2$ as follows:

$$L_2T^2D^2 \le L_2\max\left(\left\lceil\left(\frac{20\sigma}{L_2D^2}\right)^{\frac{2}{5}}\right\rceil, \left\lceil\left(\frac{10L_1}{L_2D}\right)^{\frac{1}{3}}\right\rceil\right)^2 D^2$$

$$\le L_2\left\lceil\left(\frac{20\sigma}{L_2D^2}\right)^{\frac{2}{5}}\right\rceil^2 D^2 + L_2\left\lceil\left(\frac{10L_1}{L_2D}\right)^{\frac{1}{3}}\right\rceil^2 D^2.$$

Using the fact that $\lceil x\rceil^2 \le (x+1)^2 \le 2x^2 + 2$ for any $x \ge 0$, we obtain

$$L_2 T^2 D^2 \le L_2 \left( 2 \left( \frac{20\sigma}{L_2 D^2} \right)^{\frac{4}{5}} + 2 \right) D^2 + L_2 \left( 2 \left( \frac{10L_1}{L_2 D} \right)^{\frac{2}{3}} + 2 \right) D^2$$

$$= 2L_2 \left( \frac{20\sigma}{L_2 D^2} \right)^{\frac{4}{5}} D^2 + 2L_2 \left( \frac{10L_1}{L_2 D} \right)^{\frac{2}{3}} D^2 + 4L_2 D^2$$

$$\le 22 L_2^{\frac{1}{5}} \sigma^{\frac{4}{5}} D^{\frac{2}{5}} + 10 L_1^{\frac{2}{3}} L_2^{\frac{1}{3}} D^{\frac{4}{3}} + 4L_2 D^2$$

Substituting the above inequalities into the bound in (20) yields

$$\mathbb{E}\left[ \frac{1}{K} \sum_{k=1}^{K} \|\nabla F(\bar{\mathbf{w}}^k)\| \right] \le 2 \frac{F(\mathbf{x}_0) - F^*}{DM} + \max\left( 10^{\frac{2}{3}} L_1^{\frac{2}{3}} L_2^{\frac{1}{3}} D^{\frac{4}{3}}, \frac{20 L_1 D}{M} \right)$$

$$+ \max\left( 20^{\frac{4}{5}} L_2^{\frac{1}{5}} D^{\frac{2}{5}} \sigma^{\frac{4}{5}}, \frac{20\sqrt{2}\sigma}{\sqrt{M}} \right) + 22 L_2^{\frac{1}{5}} \sigma^{\frac{4}{5}} D^{\frac{2}{5}} + 10 L_1^{\frac{2}{3}} L_2^{\frac{1}{3}} D^{\frac{4}{3}} + 4L_2 D^2.$$

Using $\max(a, b) \le a + b$, and after rounding constants, we get:

$$\mathbb{E}\left[ \frac{1}{K} \sum_{k=1}^{K} \|\nabla F(\bar{\mathbf{w}}^k)\| \right] \le 2 \frac{F(\mathbf{x}_0) - F^*}{DM} + \frac{20 L_1 D}{M} + \frac{20\sqrt{2}\sigma}{\sqrt{M}} + 33 L_2^{\frac{1}{5}} \sigma^{\frac{4}{5}} D^{\frac{2}{5}} + 15 L_1^{\frac{2}{3}} L_2^{\frac{1}{3}} D^{\frac{4}{3}} + 4L_2 D^2.$$

If we set $D = \min\left\{ \left( \frac{2(F(\mathbf{x}_0) - F^*)}{33 L_2^{1/5} \sigma^{4/5} M} \right)^{5/7}, \left( \frac{2(F(\mathbf{x}_0) - F^*)}{15 M L_1^{2/3} L_2^{1/3}} \right)^{3/7} \right\}$. Given this selection of $D$, the resulting upper bounds are as follows:

- $33 L_2^{\frac{1}{5}} D^{\frac{2}{5}} \sigma^{\frac{4}{5}} \le \frac{33^{\frac{5}{7}} 2^{\frac{2}{7}} (F(\mathbf{x}_0) - F^*)^{\frac{2}{7}}}{M^{\frac{2}{7}}} L_2^{\frac{1}{7}} \sigma^{\frac{4}{7}} \le \frac{15(F(\mathbf{x}_0) - F^*)^{\frac{2}{7}}}{M^{\frac{2}{7}}} L_2^{\frac{1}{7}} \sigma^{\frac{4}{7}};$

- $15 L_1^{\frac{2}{3}} L_2^{\frac{1}{3}} D^{\frac{4}{3}} \le \frac{15^{\frac{3}{7}} 2^{\frac{4}{7}} (F(\mathbf{x}_0) - F^*)^{\frac{4}{7}}}{M^{\frac{4}{7}}} \le \frac{5(F(\mathbf{x}_0) - F^*)^{\frac{4}{7}}}{M^{\frac{4}{7}}} L_1^{\frac{2}{7}} L_2^{\frac{1}{7}};$

- $2 \frac{F(\mathbf{x}_0) - F^*}{DM} \le 2 \frac{7.5^{\frac{3}{7}} (F(\mathbf{x}_0) - F^*)^{\frac{4}{7}}}{M^{\frac{4}{7}}} + 2 \frac{16.5^{\frac{5}{7}} (F(\mathbf{x}_0) - F^*)^{\frac{2}{7}}}{M^{\frac{2}{7}}} L_2^{\frac{1}{7}} \sigma^{\frac{4}{7}} \le \frac{5(F(\mathbf{x}_0) - F^*)^{\frac{4}{7}}}{M^{\frac{4}{7}}} L_1^{\frac{2}{7}} L_2^{\frac{1}{7}} + \frac{15(F(\mathbf{x}_0) - F^*)^{\frac{2}{7}}}{M^{\frac{2}{7}}} L_2^{\frac{1}{7}} \sigma^{\frac{4}{7}};$

- $\frac{20 L_1 D}{M} \le \frac{20 L_1^{\frac{5}{7}} (F(\mathbf{x}_0) - F^*)^{\frac{3}{7}}}{7.5^{\frac{3}{7}} L_2^{1/7} M^{\frac{10}{7}}} \le 9 \frac{L_1^{\frac{5}{7}} (F(\mathbf{x}_0) - F^*)^{\frac{3}{7}}}{L_2^{1/7} M^{\frac{10}{7}}};$

- $4 L_2 D^2 \le \frac{4 L_2^{\frac{5}{7}} (F(\mathbf{x}_0) - F^*)^{\frac{6}{7}}}{7.5^{\frac{6}{7}} M^{\frac{6}{7}} L_1^{\frac{4}{7}}} \le \frac{L_2^{\frac{5}{7}} (F(\mathbf{x}_0) - F^*)^{\frac{6}{7}}}{L_1^{\frac{4}{7}} M^{\frac{6}{7}}}.$

Substituting these inequalities into the bound yields

$$\mathbb{E}\left[ \frac{1}{K} \sum_{k=1}^{K} \|\nabla F(\bar{\mathbf{w}}^k)\| \right] \le \frac{10(F(\mathbf{x}_0) - F^*)^{\frac{4}{7}}}{M^{\frac{4}{7}}} L_1^{\frac{2}{7}} L_2^{\frac{1}{7}} + \frac{30(F(\mathbf{x}_0) - F^*)^{\frac{2}{7}}}{M^{\frac{2}{7}}} L_2^{\frac{1}{7}} \sigma^{\frac{4}{7}}$$

$$+ 9 \frac{L_1^{\frac{5}{7}} (F(\mathbf{x}_0) - F^*)^{\frac{3}{7}}}{L_2^{1/7} M^{\frac{10}{7}}} + \frac{L_2^{\frac{5}{7}} (F(\mathbf{x}_0) - F^*)^{\frac{6}{7}}}{L_1^{\frac{4}{7}} M^{\frac{6}{7}}} + \frac{20\sqrt{2}\sigma}{\sqrt{M}}$$

This final bound confirms the convergence rate stated in the main theorem, as the last three terms are of higher order and thus do not influence the rate.

## C  Proofs for Online Doubly Optimistic Gradient method with adaptive step size

In this section, we provide detailed proofs corresponding to Section 4, where the *Online Doubly Optimistic Gradient* method with an adaptive step size was introduced, along with its complexity analysis. The section is organized into several subsections, presenting useful inequalities and lemmas that build up to the proofs of the two main results from Section 4, namely Lemma 4.1 and Theorem 4.2.

## C.1 HELPFUL INEQUALITIES

**Lemma C.1.** *Let $a_i \geq 0$ $\forall i$, and that there exists at least one index $j \in [1, T]$ with $a_j > 0$. Then the following inequality holds*

$$\sqrt{\sum_{i=1}^{T} a_i} \leq \sum_{i=1}^{T} \frac{a_i}{\sqrt{\sum_{j=1}^{i} a_j}} \leq 2\sqrt{\sum_{i=1}^{T} a_i}.$$

Lemma C.1 appears in Kavis et al. (2019); Antonakopoulos et al. (2022); Levy et al. (2018); for the sake of completeness, we additionally provide its proof here.

*Proof.* We prove the lower and upper bounds separately.
**Lower bound:** We observe that

$$\sum_{i=1}^{T} a_i = \sum_{i=1}^{T} a_i \left( \frac{\sqrt{\sum_{j=1}^{i} a_j}}{\sqrt{\sum_{j=1}^{i} a_j}} \right) \leq \sum_{i=1}^{T} a_i \left( \frac{\sqrt{\sum_{j=1}^{T} a_j}}{\sqrt{\sum_{j=1}^{i} a_j}} \right),$$

where the second inequality follows from the fact that $a_i \geq 0$ $\forall i$. Dividing both sides by $\sqrt{\sum_{j=1}^{T} a_j}$ gives the desired bound.

**Upper bound:** Define $S_j = \sum_{i=1}^{j} a_i$, with the convention that $S_0 = 0$. For each $i$, it holds that

$$\frac{a_i}{2\sqrt{S_i}} \leq \frac{a_i}{\sqrt{S_i} + \sqrt{S_{i-1}}} = \frac{S_i - S_{i-1}}{\sqrt{S_i} + \sqrt{S_{i-1}}} = \sqrt{S_i} - \sqrt{S_{i-1}},$$

where the first inequality uses $S_{i-1} \leq S_i$, the first equality uses $a_i = S_i - S_{i-1}$, and the last equality follows from the difference of squares. Summing both sides of the inequality from $i = 1$ to $i = T$, yields:

$$\sum_{i=1}^{T} \frac{a_i}{2\sqrt{S_i}} \leq \sum_{i=1}^{T} \sqrt{S_i} - \sqrt{S_{i-1}} = \sqrt{S_T},$$

where the first equality follows from telescoping the series, and note that by definition $S_0 = 0$. Multiplying both sides by 2 gives the desired inequality. $\square$

**Lemma C.2.** *Let $a, b \in \mathbb{R}$, and define $M = \min\left\{(a+b)^2, a^2\right\}$. Then the following inequality holds:*
$$(a+b)^2 \leq 2M + 2b^2.$$

*Proof.* We have both $(a+b)^2 \leq 2a^2 + 2b^2$ and $(a+b)^2 \leq 2(a+b)^2 + 2b^2$. Combining these two gives us the desired inequality. $\square$

## C.2 PROOF OF LEMMA 4.1

Prior to proving Lemma 4.1, we state the following proposition, which is an immediate consequence of Lemma B.1.

**Proposition C.3.** *Consider the update in (5), with an adaptive step size $\eta_n$. Then for any $\mathbf{u}$ such that $\|\mathbf{u}\| \leq D$, it holds that*

$$\langle \mathbf{g}_{n+1}, \boldsymbol{\Delta}_{n+1} - \mathbf{u} \rangle \leq \frac{1}{2\eta_n} \|\boldsymbol{\Delta}_n - \mathbf{u}\|^2 - \frac{1}{2\eta_n} \|\boldsymbol{\Delta}_{n+1} - \mathbf{u}\|^2 + \langle \mathbf{g}_{n+1} - \mathbf{h}_{n+1}, \boldsymbol{\Delta}_{n+1} - \mathbf{u} \rangle$$

$$- \langle \mathbf{g}_n - \mathbf{h}_n, \boldsymbol{\Delta}_n - \mathbf{u} \rangle + \eta_n \|\mathbf{g}_n - \mathbf{h}_n\|^2 - \frac{1}{4\eta_n} \|\boldsymbol{\Delta}_{n+1} - \boldsymbol{\Delta}_n\|^2.$$

*Remark* C.1. To simplify the notation in the remainder of the proof, we define $\bar{\mathbf{g}}_n := \nabla F(\mathbf{w}_n)$ and $\bar{\mathbf{h}}_n := \nabla F(\mathbf{z}_{n-1})$.

We are now prepared to present the proof of Lemma 4.1.

*Proof of Lemma 4.1.* The first step in proving Lemma 4.1 is to bound the regret within the episode. Analogously to Lemma B.2, we can formulate the following lemma for the first episode using an adaptive step size.

**Lemma C.4.** *Consider the optimistic update in* (5) *with an adaptive step size $\eta_n$. Then for $k = 1$ and any $\mathbf{u}^1$ such that $\|\mathbf{u}^1\| \leq D$, we have*

$$\sum_{n=1}^{T} \langle \mathbf{g}_n, \boldsymbol{\Delta}_n - \mathbf{u}^1 \rangle \leq \frac{3D^2}{\eta_T} + \sum_{n=1}^{T} \left( \eta_n \|\mathbf{g}_n - \mathbf{h}_n\|^2 - \frac{1}{4\eta_n} \|\boldsymbol{\Delta}_n - \boldsymbol{\Delta}_{n-1}\|^2 \right).$$

The proof of this lemma is provided in Appendix C.3. Furthermore, the following lemma allows us to upper bound the sum $\sum_{n=1}^{T} \eta_n \|\mathbf{g}_n - \mathbf{h}_n\|^2$ in terms of $\sum_{n=1}^{T} \|\mathbf{g}_n - \mathbf{h}_n\|^2$, effectively decoupling the product of random variables. The proof can be found in Apendix C.3

**Lemma C.5.** *Using the definition of $\eta_n$ from* (8)*, we can derive the following bound*

$$\frac{3D}{\eta_T} + \sum_{n=1}^{T} \eta_n \|\mathbf{g}_n - \mathbf{h}_n\|^2 \leq 2\sqrt{2} \left( \frac{3D}{2\gamma} + \gamma D \right) \sqrt{\alpha + \sum_{n=1}^{T} \|\mathbf{g}_n - \mathbf{h}_n\|^2}$$

In addition, define $\mu_n = \min \left( \|\mathbf{g}_n - \mathbf{h}_n\|^2, \|\bar{\mathbf{g}}_n - \bar{\mathbf{h}}_n\|^2 \right)$. Applying Lemma C.2, we obtain the inequality $\|\mathbf{g}_n - \mathbf{h}_n\|^2 \leq 2\mu_n + 4\|\mathbf{g}_n - \bar{\mathbf{g}}_n\|^2 + 4\|\mathbf{h}_n - \bar{\mathbf{h}}_n\|^2$. This allows us to upper bound the right-hand side of the expression as follows

$$\sqrt{\alpha + \sum_{n=1}^{T} \|\mathbf{g}_n - \mathbf{h}_n\|^2} \leq 2\sqrt{\sum_{n=1}^{T} (\|\mathbf{g}_n - \bar{\mathbf{g}}_n\|^2 + \|\mathbf{h}_n - \bar{\mathbf{h}}_n\|^2)} + \sqrt{2}\sqrt{\alpha + \sum_{n=1}^{T} \mu_n}.$$

This effectively decouples the noise in $\|\mathbf{g}_n - \mathbf{h}_n\|$ into separate, more manageable components. We now focus on bounding the expectation of the first term in the preceding inequality. By applying Jensen's inequality and invoking Assumption 2.3, we obtain: $\mathbb{E}\left[ 2\sqrt{\sum_{n=1}^{T}(\|\mathbf{g}_n - \bar{\mathbf{g}}_n\|^2 + \|\mathbf{h}_n - \bar{\mathbf{h}}_n\|^2)} \right] \leq 2\sqrt{\sum_{n=1}^{T} \mathbb{E}[\|\mathbf{g}_n - \bar{\mathbf{g}}_n\|^2 + \|\mathbf{h}_n - \bar{\mathbf{h}}_n\|^2]} \leq 2\sqrt{2}\sigma\sqrt{T}$. Therefore, it only remains to upper bound

$$4 \left( \frac{3D}{2\gamma} + \gamma D \right) \sqrt{\alpha + \sum_{n=1}^{T} \mu_n} - \sum_{n=1}^{T} \frac{1}{4\eta_n} \|\boldsymbol{\Delta}_n - \boldsymbol{\Delta}_{n-1}\|^2. \tag{21}$$

Note that by the definition of $\eta_n$ in (8), we have $\eta_n \leq \frac{\gamma D}{\sqrt{\alpha + \sum_{i=1}^{n} \mu_i}}$. Also, by the definition $\hat{L}_1^* = \max_{1 \leq n \leq M} \frac{\|\mathbf{g}_n - \mathbf{h}_n\|}{\|\mathbf{w}_n - \mathbf{z}_{n-1}\|}$, we have $\frac{1}{4}\|\boldsymbol{\Delta}_n - \boldsymbol{\Delta}_{n-1}\|^2 \geq \frac{1}{(\hat{L}_1^*)^2}\|\bar{\mathbf{g}}_n - \bar{\mathbf{h}}_n\|^2 \geq \frac{1}{(\hat{L}_1^*)^2}\mu_n$. Hence, we obtain that $\sum_{n=1}^{T} \frac{1}{4\eta_n}\|\boldsymbol{\Delta}_n - \boldsymbol{\Delta}_{n-1}\|^2 \geq \sum_{n=1}^{T} \frac{\sqrt{\alpha + \sum_{i=1}^{n} \mu_i}}{\gamma D (\hat{L}_1^*)^2}\mu_n$. Hence, the term in (21) can be upper bounded by $4(\frac{3D}{2\gamma} + \gamma D)\sqrt{\alpha + \sum_{n=1}^{T} \mu_n} - \sum_{n=1}^{T} \frac{\sqrt{\alpha + \sum_{i=1}^{n} \mu_i}}{\gamma D (\hat{L}_1^*)^2}\mu_n$, which in turn is bounded in the following lemma. Its proof can be found in Appendix C.5

**Lemma C.6.** *Given the previously defined quantities $\mu_n$ and $\hat{L}_1^*$, we have the following*

$$4 \left( \frac{3D}{2\gamma} + \gamma D \right) \sqrt{\alpha + \sum_{n=1}^{T} \mu_n} - \sum_{n=1}^{T} \frac{\sqrt{\alpha + \sum_{i=1}^{n} \mu_i}}{\gamma D (\hat{L}_1^*)^2}\mu_n \leq 16 \left( \frac{3}{2\gamma} + \gamma \right)^{\frac{3}{2}} \hat{L}_1^* \gamma^{\frac{1}{2}} D^2,$$

Combining all the pieces together yields

$$\sum_{n=1}^{T} \mathbb{E}\left[ \langle \mathbf{g}_n, \boldsymbol{\Delta}_n - \mathbf{u}^1 \rangle \right] \leq 8 \left( \frac{3}{\gamma} + \gamma \right) D\sqrt{T}\sigma + 16 \left( \frac{3}{2\gamma} + \gamma \right)^{\frac{3}{2}} \gamma^{\frac{1}{2}} D^2,$$

and this completes the proof. □

## C.3 PROOF OF LEMMA C.4

Setting $\mathbf{u} = \mathbf{u}^1$, we sum the inequality from Proposition C.3 over $n = 1$ to $n = T - 1$

$$
\sum_{n=2}^{T} \langle \mathbf{g}_n, \boldsymbol{\Delta}_n - \mathbf{u}^1 \rangle \leq \langle \mathbf{g}_T - \mathbf{h}_T, \boldsymbol{\Delta}_T - \mathbf{u}^1 \rangle - \langle \mathbf{g}_1 - \mathbf{h}_1, \boldsymbol{\Delta}_1 - \mathbf{u}^1 \rangle + \sum_{n=1}^{T-1} \eta_n \|\mathbf{g}_n - \mathbf{h}_n\|^2
$$

$$
+ \sum_{n=1}^{T-1} \left( \frac{1}{2\eta_{n+1}} - \frac{1}{2\eta_n} \right) \|\boldsymbol{\Delta}_{n+1} - \mathbf{u}^1\|^2 + \frac{1}{2\eta_1} \|\boldsymbol{\Delta}_1 - \mathbf{u}^1\|^2 - \frac{1}{2\eta_T} \|\boldsymbol{\Delta}_T - \mathbf{u}^1\|^2
$$

$$
- \sum_{n=1}^{T-1} \frac{1}{4\eta_n} \|\boldsymbol{\Delta}_{n+1} - \boldsymbol{\Delta}_n\|^2 .
$$

First, note that by the definitions of $\mathbf{h}_1$ and $\boldsymbol{\Delta}_1$, we have $\boldsymbol{\Delta}_1 = \arg\min_{\|\boldsymbol{\Delta}\| \leq D} \langle \mathbf{h}_1, \boldsymbol{\Delta} \rangle$. This implies that $\langle \mathbf{h}_1, \boldsymbol{\Delta}_1 - \mathbf{u}^1 \rangle \leq 0$. Therefore

$$
\sum_{n=1}^{T} \langle \mathbf{g}_n, \boldsymbol{\Delta}_n - \mathbf{u}^1 \rangle \leq \langle \mathbf{g}_T - \mathbf{h}_T, \boldsymbol{\Delta}_T - \mathbf{u}^1 \rangle + \sum_{n=1}^{T-1} \eta_n \|\mathbf{g}_n - \mathbf{h}_n\|^2 + \sum_{n=1}^{T-1} \left( \frac{1}{2\eta_{n+1}} - \frac{1}{2\eta_n} \right) \|\boldsymbol{\Delta}_{n+1} - \mathbf{u}^1\|^2
$$

$$
+ \frac{1}{2\eta_1} \|\boldsymbol{\Delta}_1 - \mathbf{u}^1\|^2 - \frac{1}{2\eta_T} \|\boldsymbol{\Delta}_T - \mathbf{u}^1\|^2 - \sum_{n=1}^{T-1} \frac{1}{4\eta_n} \|\boldsymbol{\Delta}_{n+1} - \boldsymbol{\Delta}_n\|^2 .
$$

Additionally, applying the Cauchy-Schwarz and Young's inequalities, we can bound

$$
\langle \mathbf{g}_T - \mathbf{h}_T, \boldsymbol{\Delta}_T - \mathbf{u}^1 \rangle \leq \|\mathbf{g}_T - \mathbf{h}_T\| \|\boldsymbol{\Delta}_T - \mathbf{u}^1\|
$$

$$
\leq \frac{\eta_T}{2} \|\mathbf{g}_T - \mathbf{h}_T\|^2 + \frac{1}{2\eta_T} \|\boldsymbol{\Delta}_T - \mathbf{u}^1\|^2 .
$$

Combining all the inequalities, we obtain

$$
\sum_{n=1}^{T} \langle \mathbf{g}_n, \boldsymbol{\Delta}_n - \mathbf{u}^1 \rangle \leq \frac{\eta_T}{2} \|\mathbf{g}_T - \mathbf{h}_T\|^2 + \sum_{n=1}^{T-1} \left( \eta_n \|\mathbf{g}_n - \mathbf{h}_n\|^2 - \frac{1}{4\eta_n} \|\boldsymbol{\Delta}_{n+1} - \boldsymbol{\Delta}_n\|^2 \right)
$$

$$
+ \sum_{n=1}^{T-1} \left( \frac{1}{2\eta_{n+1}} - \frac{1}{2\eta_n} \right) \|\boldsymbol{\Delta}_{n+1} - \mathbf{u}^1\|^2 + \frac{1}{2\eta_1} \|\boldsymbol{\Delta}_1 - \mathbf{u}^1\|^2 .
$$

As $\eta_n$ is a decreasing sequence, by definition (8), we have $\left( \frac{1}{2\eta_{n+1}} - \frac{1}{2\eta_n} \right) \geq 0$. Additionally, $\|\boldsymbol{\Delta}_n\| \leq D$ and $\|\mathbf{u}^1\| \leq D$. Therefore, we have $\sum_{n=1}^{T-1} \left( \frac{1}{2\eta_{n+1}} - \frac{1}{2\eta_n} \right) \|\boldsymbol{\Delta}_{n+1} - \mathbf{u}^1\|^2 + \frac{1}{2\eta_1} \|\boldsymbol{\Delta}_1 - \mathbf{u}^1\|^2 \leq 4D^2 \sum_{n=1}^{T-1} \left( \frac{1}{2\eta_{n+1}} - \frac{1}{2\eta_n} \right) + \frac{4D^2}{2\eta_1} = \frac{2D^2}{\eta_T}$. Hence

$$
\sum_{n=1}^{T} \langle \mathbf{g}_n, \boldsymbol{\Delta}_n - \mathbf{u}^1 \rangle \leq \frac{\eta_T}{2} \|\mathbf{g}_T - \mathbf{h}_T\|^2 + \sum_{n=1}^{T-1} \left( \eta_n \|\mathbf{g}_n - \mathbf{h}_n\|^2 - \frac{1}{4\eta_n} \|\boldsymbol{\Delta}_{n+1} - \boldsymbol{\Delta}_n\|^2 \right) + \frac{2D^2}{\eta_T}
$$

$$
= \frac{2D^2}{\eta_T} + \frac{\eta_T}{2} \|\mathbf{g}_T - \mathbf{h}_T\|^2 + \sum_{n=2}^{T-1} \left( \eta_n \|\mathbf{g}_n - \mathbf{h}_n\|^2 - \frac{1}{4\eta_{n-1}} \|\boldsymbol{\Delta}_n - \boldsymbol{\Delta}_{n-1}\|^2 \right)
$$

$$
- \frac{1}{4\eta_{T-1}} \|\boldsymbol{\Delta}_T - \boldsymbol{\Delta}_{T-1}\|^2 + \eta_1 \|\mathbf{g}_1 - \mathbf{h}_1\|^2
$$

Note that $\frac{\eta_T}{2}\|\mathbf{g}_T - \mathbf{h}_T\|^2 \le \eta_T \|\mathbf{g}_T - \mathbf{h}_T\|^2$. Therefore, we can state the following upper bound

$$\sum_{n=1}^{T}\langle \mathbf{g}_n, \boldsymbol{\Delta}_n - \mathbf{u}^1 \rangle \le \frac{2D^2}{\eta_T} + \sum_{n=2}^{T}\left( \eta_n \|\mathbf{g}_n - \mathbf{h}_n\|^2 - \frac{1}{4\eta_{n-1}}\|\boldsymbol{\Delta}_n - \boldsymbol{\Delta}_{n-1}\|^2 \right) + \eta_1\|\mathbf{g}_1 - \mathbf{h}_1\|^2$$

$$= \frac{2D^2}{\eta_T} + \sum_{n=2}^{T}\left( \eta_n \|\mathbf{g}_n - \mathbf{h}_n\|^2 - \frac{1}{4\eta_n}\|\boldsymbol{\Delta}_n - \boldsymbol{\Delta}_{n-1}\|^2 \right) + \eta_1\|\mathbf{g}_1 - \mathbf{h}_1\|^2$$

$$+ \frac{1}{4}\sum_{n=2}^{T}\left( \frac{1}{\eta_n} - \frac{1}{\eta_{n-1}} \right)\|\boldsymbol{\Delta}_n - \boldsymbol{\Delta}_{n-1}\|^2$$

$$\le \frac{2D^2}{\eta_T} + \sum_{n=2}^{T}\left( \eta_n \|\mathbf{g}_n - \mathbf{h}_n\|^2 - \frac{1}{4\eta_n}\|\boldsymbol{\Delta}_n - \boldsymbol{\Delta}_{n-1}\|^2 \right) + \eta_1\|\mathbf{g}_1 - \mathbf{h}_1\|^2$$

$$+ \frac{D^2}{\eta_T} - \frac{D^2}{\eta_1}$$

Here, the first equality follows by adding and subtracting the term $\sum_{n=2}^{T}\frac{1}{4\eta_n}\|\boldsymbol{\Delta}_n - \boldsymbol{\Delta}_{n-1}\|^2$, while the final inequality uses the facts that $\left( \frac{1}{\eta_n} - \frac{1}{\eta_{n-1}} \right) \ge 0$ and $\|\boldsymbol{\Delta}_n - \boldsymbol{\Delta}_{n-1}\|^2 \le 4D^2$. Moreover, we observe that $-\frac{D^2}{\eta_1} \le -\frac{1}{4\eta_1}\|\boldsymbol{\Delta}_1 - \boldsymbol{\Delta}_0\|^2$. Therefore

$$\sum_{n=1}^{T}\langle \mathbf{g}_n, \boldsymbol{\Delta}_n - \mathbf{u}^1 \rangle \le \frac{3D^2}{\eta_T} + \sum_{n=1}^{T}\left( \eta_n \|\mathbf{g}_n - \mathbf{h}_n\|^2 - \frac{1}{4\eta_n}\|\boldsymbol{\Delta}_n - \boldsymbol{\Delta}_{n-1}\|^2 \right).$$

This completes the proof.

### C.4 Proof of Lemma C.5

Using $\eta_n$ as defined in (8), and noting that we are analyzing the first episode (so $n \bmod T$ simplifies to $n$), we have

$$\frac{3D}{\eta_T} + \sum_{n=1}^{T}\eta_n \|\mathbf{g}_n - \mathbf{h}_n\|^2 \le \frac{3D}{\gamma}\sqrt{\alpha + \sum_{i=1}^{T}\|\mathbf{g}_i - \mathbf{h}_i\|^2} + \sum_{n=1}^{T}\frac{\gamma D \|\mathbf{g}_n - \mathbf{h}_n\|^2}{\sqrt{\alpha + \sum_{i=1}^{n}\|\mathbf{g}_i - \mathbf{h}_i\|^2}}.$$

Applying the inequality $\sqrt{a + b} \le \sqrt{a} + \sqrt{b}$ for $a, b \ge 0$, we obtain

$$\frac{3D}{\eta_T} + \sum_{n=1}^{T}\eta_n \|\mathbf{g}_n - \mathbf{h}_n\|^2 \le \frac{3D}{\gamma}\sqrt{\alpha} + \frac{3D}{\gamma}\sqrt{\sum_{i=1}^{T}\|\mathbf{g}_i - \mathbf{h}_i\|^2} + \sum_{n=1}^{T}\frac{\gamma D \|\mathbf{g}_n - \mathbf{h}_n\|^2}{\sqrt{\sum_{i=1}^{n}\|\mathbf{g}_i - \mathbf{h}_i\|^2}}.$$

Applying the upper bound corresponding to the lower bound from Lemma C.1, we get

$$\frac{3D}{\eta_T} + \sum_{n=1}^{T}\eta_n \|\mathbf{g}_n - \mathbf{h}_n\|^2 \le \frac{3D}{\gamma}\sqrt{\alpha} + \frac{3D}{\gamma}\sqrt{\sum_{i=1}^{T}\|\mathbf{g}_i - \mathbf{h}_i\|^2} + 2\gamma D\sqrt{\sum_{i=1}^{T}\|\mathbf{g}_i - \mathbf{h}_i\|^2}.$$

Utilizing the fact that $(\sqrt{a} + \sqrt{b})^2 \le 2(a + b)$ leads to

$$\frac{3D}{\eta_T} + \sum_{n=1}^{T}\eta_n \|\mathbf{g}_n - \mathbf{h}_n\|^2 \le 2\sqrt{2}\left( \frac{3D}{2\gamma} + \gamma D \right)\sqrt{\alpha + \sum_{n=1}^{T}\|\mathbf{g}_n - \mathbf{h}_n\|^2}.$$

### C.5 PROOF OF LEMMA C.6

*Proof.* Applying Lemma C.1, we obtain

$$
4\left(\frac{3D}{2\gamma} + \gamma D\right)\sqrt{\alpha + \sum_{n=1}^{T}\mu_n} - \sum_{n=1}^{T}\frac{\sqrt{\alpha + \sum_{i=1}^{n}\mu_i}}{\gamma D(\hat{L}_1^*)^2}\mu_n
$$

$$
\leq 4\left(\frac{3D}{2\gamma} + \gamma D\right)\sqrt{\alpha} + 4\left(\frac{3D}{2\gamma} + \gamma D\right)\frac{\sum_{n=1}^{T}\mu_n}{\sqrt{\alpha + \sum_{i=1}^{n}\mu_i}} - \sum_{n=1}^{T}\frac{\sqrt{\alpha + \sum_{i=1}^{n}\mu_i}}{(\hat{L}_1^*)^2\gamma D}\mu_n.
$$

Reorganizing the terms, we further have

$$
4\left(\frac{3D}{2\gamma} + \gamma D\right)\sqrt{\alpha + \sum_{n=1}^{T}\mu_n} - \sum_{n=1}^{T}\frac{\sqrt{\alpha + \sum_{i=1}^{n}\mu_i}}{\gamma D(\hat{L}_1^*)^2}\mu_n
$$

$$
\leq 4\left(\frac{3D}{2\gamma} + \gamma D\right)\sqrt{\alpha} + \sum_{n=1}^{T}\left(4\left(\frac{3D}{2\gamma} + \gamma D\right) - \frac{\alpha + \sum_{i=1}^{n}\mu_i}{(\hat{L}_1^*)^2\gamma D}\right)\frac{\mu_n}{\sqrt{\alpha + \sum_{i=1}^{n}\mu_i}}. \tag{22}
$$

Observe that the quantity $\left(4\left(\frac{3D}{2\gamma} + \gamma D\right) - \frac{\alpha + \sum_{i=1}^{n}\mu_i}{4(\hat{L}_1^*)^2\gamma D}\right)$ defines a decreasing sequence with respect to $n$. Define $T^*$ as the smallest integer such that for all $n > T^*$, the following holds

$$
4\left(\frac{3D}{2\gamma} + \gamma D\right) - \frac{\alpha + \sum_{i=1}^{n}\mu_i}{(\hat{L}_1^*)^2\gamma D} \leq 0
$$

*Remark* C.2. For sufficiently small $\alpha$, we can ensure that the following two inequalities hold, where $T^*$ is an integer between $1$ and $T$:

Then, the following two scenarios hold, depending on the value of $n$

$$
4\left(\frac{3D}{2\gamma} + \gamma D\right)(\hat{L}_1^*)^2\gamma D \leq \alpha + \sum_{i=1}^{n}\mu_i \qquad \forall\, n > T^*
$$

$$
4\left(\frac{3D}{2\gamma} + \gamma D\right)(\hat{L}_1^*)^2\gamma D \geq \alpha + \sum_{i=1}^{n}\mu_i \qquad \forall\, n \leq T^* \tag{23}
$$

Returning to equation (22), with this definition of $T^*$, we have

$$
4\left(\frac{3D}{2\gamma} + \gamma D\right)\sqrt{\alpha + \sum_{n=1}^{T}\mu_n} - \sum_{n=1}^{T}\frac{\sqrt{\alpha + \sum_{i=1}^{n}\mu_i}}{\gamma D(\hat{L}_1^*)^2}\mu_n \leq 4\left(\frac{3D}{2\gamma} + \gamma D\right)\sqrt{\alpha}
$$

$$
+ \sum_{n=1}^{T^*}\left(4\left(\frac{3D}{2\gamma} + \gamma D\right) - \frac{\alpha + \sum_{i=1}^{n}\mu_i}{(\hat{L}_1^*)^2\gamma D}\right)\frac{\mu_n}{\sqrt{\alpha + \sum_{i=1}^{n}\mu_i}}.
$$

By discarding the negative terms, we obtain

$$
4\left(\frac{3D}{2\gamma} + \gamma D\right)\sqrt{\alpha + \sum_{n=1}^{T}\mu_n} - \sum_{n=1}^{T}\frac{\sqrt{\alpha + \sum_{i=1}^{n}\mu_i}}{\gamma D(\hat{L}_1^*)^2}\mu_n \leq 4\left(\frac{3D}{2\gamma} + \gamma D\right)\sqrt{\alpha}
$$

$$
+ \sum_{n=1}^{T^*}4\left(\frac{3D}{2\gamma} + \gamma D\right)\frac{\mu_n}{\sqrt{\alpha + \sum_{i=1}^{n}\mu_i}}.
$$

Observing that $\frac{\mu_n}{\sqrt{\alpha + \sum_{i=1}^n \mu_i}} \leq \frac{\mu_n}{\sqrt{\sum_{i=1}^n \mu_i}}$, and applying Lemma C.1 to this sequence, we obtain

$$4 \left( \frac{3D}{2\gamma} + \gamma D \right) \sqrt{\alpha + \sum_{n=1}^{T} \mu_n} - \sum_{n=1}^{T} \frac{\sqrt{\alpha + \sum_{i=1}^n \mu_i}}{\gamma D (\hat{L}_1^*)^2} \mu_n \leq 4 \left( \frac{3D}{2\gamma} + \gamma D \right) \sqrt{\alpha}$$
$$+ 8 \left( \frac{3D}{2\gamma} + \gamma D \right) \sqrt{\sum_{n=1}^{T^*} \mu_n}.$$

Combining terms

$$4 \left( \frac{3D}{2\gamma} + \gamma D \right) \sqrt{\alpha + \sum_{n=1}^{T} \mu_n} - \sum_{n=1}^{T} \frac{\sqrt{\alpha + \sum_{i=1}^n \mu_i}}{\gamma D (\hat{L}_1^*)^2} \mu_n \leq 8 \left( \frac{3D}{2\gamma} + \gamma D \right) \sqrt{\alpha + \sum_{n=1}^{T^*} \mu_n}.$$

Invoking the upper bound for $T^*$ from (23),

$$4 \left( \frac{3D}{2\gamma} + \gamma D \right) \sqrt{\alpha + \sum_{n=1}^{T} \mu_n} - \sum_{n=1}^{T} \frac{\sqrt{\alpha + \sum_{i=1}^n \mu_i}}{\gamma D (\hat{L}_1^*)^2} \mu_n \leq 8 \left( \frac{3D}{2\gamma} + \gamma D \right) \sqrt{4 \left( \frac{3D}{2\gamma} + \gamma D \right) (\hat{L}_1^*)^2 \gamma D}$$
$$= 16 \left( \frac{3}{2\gamma} + \gamma \right)^{\frac{3}{2}} \hat{L}_1^* \gamma^{\frac{1}{2}} D^2.$$

This concludes the proof. $\qquad\square$

## C.6 PROOF OF THEOREM 4.2

Note that the algorithm restarts after every $T$ iterations. Therefore, the bounds proven for the first episode apply to each subsequent episode, and as a corollary of Lemma 4.1 we obtain
$\mathbb{E} \left[ \text{Reg}_T(\mathbf{u}^1, \dots, \mathbf{u}^K) \right] \leq 8 \left( \frac{3}{2\gamma} + \gamma \right) DK\sqrt{T}\sigma + 16 \left( \frac{3}{2\gamma} + \gamma \right)^{\frac{3}{2}} \hat{L}_1^* \gamma^{\frac{1}{2}} KD^2$. Together with Proposition 2.1, we can show that:

$$\frac{1}{K} \sum_{k=1}^{K} \|\nabla F(\bar{\mathbf{w}}^k)\| \leq \frac{F(\mathbf{x}_0) - F^*}{DM} + 8C_1 \frac{\sigma}{\sqrt{T}} + 16C_1^{\frac{3}{2}} \hat{L}_1^* \gamma^{\frac{1}{2}} \frac{D}{T} + \frac{L_2}{48} D^2 + \frac{L_2}{2} T^2 D^2 + \frac{\sigma}{\sqrt{T}},$$

where $C_1 = \left( \frac{3}{2\gamma} + \gamma \right)$, also define $C_2 = \left( \frac{12}{\gamma} + 8\gamma + 1 \right)$.

Equivalently

$$\frac{1}{K} \sum_{k=1}^{K} \|\nabla F(\bar{\mathbf{w}}^k)\| \leq \frac{F(\mathbf{x}_0) - F^*}{DKT} + C_2 \frac{\sigma}{\sqrt{T}} + 16C_1^{\frac{3}{2}} \hat{L}_1^* \gamma^{\frac{1}{2}} \frac{D}{T} + L_2 T^2 D^2. \qquad (24)$$

Based on the upper bound in (24), the hyperparameters $D, K, T$ must be chosen under the constraint $KT \leq M$, where $M$ denotese the total iteration budget and $K, T \in \mathbb{N}$. To obtain the sharpest bound, we will first determine $T$ and $K$, and then specify $D$.

If we set $T = \min \left( \max \left( \left\lceil \left( \frac{C_2 \sigma}{L_2 D^2} \right)^{\frac{2}{5}} \right\rceil, \left\lceil \left( 16C_1^{\frac{3}{2}} \frac{\hat{L}_1^* \gamma^{\frac{1}{2}}}{L_2 D} \right)^{\frac{1}{3}} \right\rceil \right), \frac{M}{2} \right)$ and $K = \lfloor \frac{M}{T} \rfloor$, we can bound the first three terms of the inequality (24) as follows:

- $KT \geq (\frac{M}{T} - 1)T \geq M - T \geq M/2$ which implies $\frac{F(\mathbf{x}_0) - F^*}{DKT} \leq 2 \frac{F(\mathbf{x}_0) - F^*}{DM}$;
- $16C_1^{\frac{3}{2}} \hat{L}_1^* \gamma^{\frac{1}{2}} \frac{D}{T} \quad \leq \quad \max \left( 16^{\frac{2}{3}} L_2^{\frac{1}{3}} (\hat{L}_1^*)^{\frac{2}{3}} \gamma^{\frac{1}{3}} C_1 D^{\frac{4}{3}}, 32C_1^{\frac{3}{2}} \hat{L}_1^* \gamma^{\frac{1}{2}} \frac{D}{M} \right) \quad \leq$
  $\max \left( 7L_2^{\frac{1}{3}} (\hat{L}_1^*)^{\frac{2}{3}} \gamma^{\frac{1}{3}} C_1 D^{\frac{4}{3}}, 32C_1^{\frac{3}{2}} \hat{L}_1^* \gamma^{\frac{1}{2}} \frac{D}{M} \right)$;
- $C_2 \frac{\sigma}{\sqrt{T}} \leq \max \left( C_2^{\frac{4}{5}} L_2^{\frac{1}{5}} D^{\frac{2}{5}} \sigma^{\frac{4}{5}}, C_2 \sqrt{2} \frac{\sigma}{\sqrt{M}} \right)$.

We upper bound $L_2 T^2 D^2$ as follows:

$$L_2 T^2 D^2 \leq L_2 \max\left(\left\lceil\left(\frac{C_2\sigma}{L_2 D^2}\right)^{\frac{2}{5}}\right\rceil, \left\lceil\left(16 C_1^{\frac{3}{2}}\frac{\hat{L}_1^*\gamma^{\frac{1}{2}}}{L_2 D}\right)^{\frac{1}{3}}\right\rceil\right)^2 D^2$$

$$\leq L_2 \left\lceil\left(\frac{C_2\sigma}{L_2 D^2}\right)^{\frac{2}{5}}\right\rceil^2 D^2 + L_2 \left\lceil\left(16 C_1^{\frac{3}{2}}\frac{\hat{L}_1^*\gamma^{\frac{1}{2}}}{L_2 D}\right)^{\frac{1}{3}}\right\rceil^2 D^2.$$

Using the fact that $\lceil x\rceil^2 \leq (x+1)^2 \leq 2x^2 + 2$ for any $x \geq 0$, we obtain

$$L_2 T^2 D^2 \leq L_2\left(2\left(\frac{C_2\sigma}{L_2 D^2}\right)^{\frac{4}{5}} + 2\right)D^2 + L_2\left(2\left(16 C_1^{\frac{3}{2}}\frac{\hat{L}_1^*\gamma^{\frac{1}{2}}}{L_2 D}\right)^{\frac{2}{3}} + 2\right)D^2$$

$$= 2L_2\left(\frac{C_2\sigma}{L_2 D^2}\right)^{\frac{4}{5}}D^2 + 2L_2\left(16 C_1^{\frac{3}{2}}\frac{\hat{L}_1^*\gamma^{\frac{1}{2}}}{L_2 D}\right)^{\frac{2}{3}}D^2 + 4L_2 D^2$$

$$\leq 2C_2^{\frac{4}{5}}L_2^{\frac{1}{5}}\sigma^{\frac{4}{5}}D^{\frac{2}{5}} + 13 C_1(\hat{L}_1^*)^{\frac{2}{3}}L_2^{\frac{1}{3}}D^{\frac{4}{3}} + 4L_2 D^2.$$

Substituting these values into (24) yields

$$\frac{1}{K}\sum_{k=1}^{K}\|\nabla F(\bar{\mathbf{w}}^k)\| \leq 2\frac{F(\mathbf{x}_0) - F^*}{DM} + \max\left(7 L_2^{\frac{1}{3}}(\hat{L}_1^*)^{\frac{2}{3}}\gamma^{\frac{1}{3}}C_1 D^{\frac{4}{3}}, 32 C_1^{\frac{3}{2}}\hat{L}_1^*\gamma^{\frac{1}{2}}\frac{D}{M}\right)$$

$$+ \max\left(C_2^{\frac{4}{5}}L_2^{\frac{1}{5}}D^{\frac{2}{5}}\sigma^{\frac{4}{5}}, C_2\sqrt{2}\frac{\sigma}{\sqrt{M}}\right) + 2C_2^{\frac{4}{5}}L_2^{\frac{1}{5}}\sigma^{\frac{4}{5}}D^{\frac{2}{5}} + 13 C_1(\hat{L}_1^*)^{\frac{2}{3}}L_2^{\frac{1}{3}}D^{\frac{4}{3}}$$

$$+ 4L_2 D^2.$$

Using $\max(a,b) \leq a + b$, and after rounding constants, we get:

$$\frac{1}{K}\sum_{k=1}^{K}\|\nabla F(\bar{\mathbf{w}}^k)\| \leq 2\frac{F(\mathbf{x}_0) - F^*}{DM} + 32 C_1^{\frac{3}{2}}\hat{L}_1^*\gamma^{\frac{1}{2}}\frac{D}{M} + C_2\sqrt{2}\frac{\sigma}{\sqrt{M}}$$

$$+ 3C_2^{\frac{4}{5}}L_2^{\frac{1}{5}}\sigma^{\frac{4}{5}}D^{\frac{2}{5}} + 20 C_1(\hat{L}_1^*)^{\frac{2}{3}}L_2^{\frac{1}{3}}D^{\frac{4}{3}} + 4L_2 D^2.$$

Setting $D = \min\left\{\left(2\frac{F(\mathbf{x}_0)-F^*}{3MC_2^{\frac{4}{5}}L_2^{\frac{1}{5}}\sigma^{\frac{4}{5}}}\right)^{\frac{5}{7}}, \left(\frac{F(\mathbf{x}_0)-F^*}{10M(\hat{L}_1^*)^{\frac{2}{3}}L_2^{\frac{1}{3}}\gamma^{\frac{1}{3}}C_1}\right)^{\frac{3}{7}}\right\}$ leads to the following upper bounds:

- $3C_2^{\frac{4}{5}}L_2^{\frac{1}{5}}D^{\frac{2}{5}}\sigma^{\frac{4}{5}} \leq 3^{\frac{5}{7}}2^{\frac{2}{7}}\frac{(F(\mathbf{x}_0)-F^*)^{\frac{2}{7}}}{M^{\frac{2}{7}}}C_2^{\frac{4}{7}}L_2^{\frac{1}{7}}\sigma^{\frac{4}{7}} \leq 3\frac{(F(\mathbf{x}_0)-F^*)^{\frac{2}{7}}}{M^{\frac{2}{7}}}C_2^{\frac{4}{7}}L_2^{\frac{1}{7}}\sigma^{\frac{4}{7}};$

- $20 L_2^{\frac{1}{3}}(\hat{L}_1^*)^{\frac{2}{3}}\gamma^{\frac{1}{3}}C_1 D^{\frac{4}{3}} \leq 20\gamma^{\frac{1}{7}}C_1^{\frac{3}{7}}(\hat{L}_1^*)^{\frac{2}{7}}L_2^{\frac{1}{7}}\frac{(F(\mathbf{x}_0)-F^*)^{\frac{4}{7}}}{10^{\frac{4}{7}}M^{\frac{4}{7}}} \leq 6\gamma^{\frac{1}{7}}C_1^{\frac{3}{7}}(\hat{L}_1^*)^{\frac{2}{7}}L_2^{\frac{1}{7}}\frac{(F(\mathbf{x}_0)-F^*)^{\frac{4}{7}}}{M^{\frac{4}{7}}};$

- $2\frac{F(\mathbf{x}_0)-F^*}{DM} \leq 2 \cdot 10^{\frac{3}{7}}\gamma^{\frac{1}{7}}C_1^{\frac{3}{7}}(\hat{L}_1^*)^{\frac{2}{7}}L_2^{\frac{1}{7}}\frac{(F(\mathbf{x}_0)-F^*)^{\frac{4}{7}}}{M^{\frac{4}{7}}} + 2 \cdot 1.5^{\frac{5}{7}}\frac{(F(\mathbf{x}_0)-F^*)^{\frac{2}{7}}}{M^{\frac{2}{7}}}C_2^{\frac{4}{7}}L_2^{\frac{1}{7}}\sigma^{\frac{4}{7}} \leq 6\gamma^{\frac{1}{7}}C_1^{\frac{3}{7}}(\hat{L}_1^*)^{\frac{2}{7}}L_2^{\frac{1}{7}}\frac{(F(\mathbf{x}_0)-F^*)^{\frac{4}{7}}}{M^{\frac{4}{7}}} + 3\frac{(F(\mathbf{x}_0)-F^*)^{\frac{2}{7}}}{M^{\frac{2}{7}}}C_2^{\frac{4}{7}}L_2^{\frac{1}{7}}\sigma^{\frac{4}{7}};$

- $32 C_1^{\frac{3}{2}}\hat{L}_1^*\gamma^{\frac{1}{2}}\frac{D}{M} \leq \frac{32}{10^{\frac{3}{7}}}\frac{C_1^{\frac{15}{14}}(\hat{L}_1^*)^{\frac{5}{7}}\gamma^{\frac{5}{14}}(F(\mathbf{x}_0)-F^*)^{\frac{3}{7}}}{M^{\frac{10}{7}}L_2^{\frac{1}{7}}} \leq 12\frac{C_1^{\frac{15}{14}}(\hat{L}_1^*)^{\frac{5}{7}}\gamma^{\frac{5}{14}}(F(\mathbf{x}_0)-F^*)^{\frac{3}{7}}}{M^{\frac{10}{7}}L_2^{\frac{1}{7}}};$

- $4L_2 D^2 \leq \dfrac{4}{10^{\frac{6}{7}}} \dfrac{L_2^{\frac{5}{7}} (F(\mathbf{x}_0) - F^*)^{\frac{6}{7}}}{M^{\frac{6}{7}} (\hat{L}_1^*)^{\frac{4}{7}} \gamma^{\frac{2}{7}} C_1^{\frac{6}{7}}} \leq \dfrac{L_2^{\frac{5}{7}} (F(\mathbf{x}_0) - F^*)^{\frac{6}{7}}}{M^{\frac{6}{7}} (\hat{L}_1^*)^{\frac{4}{7}} \gamma^{\frac{2}{7}} C_1^{\frac{6}{7}}}.$

Finally, substituting this value into the bound yields

$$\frac{1}{K} \sum_{k=1}^{K} \|\nabla F(\bar{\mathbf{w}}^k)\| \leq 12 \gamma^{\frac{1}{7}} C_1^{\frac{3}{7}} (\hat{L}_1^*)^{\frac{2}{7}} L_2^{\frac{1}{7}} \frac{(F(\mathbf{x}_0) - F^*)^{\frac{4}{7}}}{M^{\frac{4}{7}}} + 6 \frac{(F(\mathbf{x}_0) - F^*)^{\frac{2}{7}}}{M^{\frac{2}{7}}} C_2^{\frac{4}{7}} L_2^{\frac{1}{7}} \sigma^{\frac{4}{7}}$$

$$+ 12 \frac{C_1^{\frac{15}{14}} (\hat{L}_1^*)^{\frac{5}{7}} \gamma^{\frac{5}{14}} (F(\mathbf{x}_0) - F^*)^{\frac{3}{7}}}{M^{\frac{10}{7}} L_2^{\frac{1}{7}}} + \frac{L_2^{\frac{5}{7}} (F(\mathbf{x}_0) - F^*)^{\frac{6}{7}}}{M^{\frac{6}{7}} (\hat{L}_1^*)^{\frac{4}{7}} \gamma^{\frac{2}{7}} C_1^{\frac{6}{7}}} + C_2 \sqrt{2} \frac{\sigma}{\sqrt{M}}.$$

This final bound confirms the convergence rate stated in the main theorem, as the last three terms are of higher order and thus do not influence the rate.

# D USE OF LARGE LANGUAGE MODELS

Large Language models were used lightly to refine wording and correct grammatical errors in parts of the paper.

