# OpenReview forum: "Improving Online-to-Nonconvex Conversion for Smooth Optimization via Double Optimism"
_ICLR.cc/2026/Conference — ICLR 2026 Poster_

### Official Review · Reviewer_miQo · 2025-10-23

**Soundness:** 3
**Presentation:** 3
**Contribution:** 2
**Rating:** 8
**Confidence:** 3

**Summary:**

This paper proposes a novel method to improve the recently proposed online-to-nonconvex conversion framework (Cutkosky et al., 2023) under gradient and Hessian smoothness assumptions. Two levels of optimism are used when solving the online learning problem in (Cutkosky et al., 2023). The algorithm achieves the complexity of $\mathcal{O}(\epsilon^{-1.75} + \sigma^2 \epsilon^{-3.5})$, which recovers the best-known rates under both deterministic and stochastic settings. An adaptive variant of the method is also proposed.

**Strengths:**

The article is well-written, with the authors clearly articulating their ideas. The derivations presented in the paper lead naturally to the proposed algorithm, making it easy for readers to follow the authors' thought process. Furthermore, the final algorithm can be implemented with a single loop, and each iteration only requires a single batch, suggesting that it could be effective in practical applications.  The theoretical improvements over (Cutkosky et al., 2023) are also sound: improving a logarithmic factor in the deterministic case,  removing a stronger assumption in the stochastic case, achieved with a unified algorithm that can be extended to adaptive optimization.

**Weaknesses:**

1. The main results of this paper become relatively straightforward given the inspiring recent work (Jiang et al., 2025), and the idea of double optimism has also appeared in that work, and that work also remarked explicitly that a $\mathcal{O}(\epsilon^{-1.75})$ complexity is achievable.  Although the results in stochastic problems and adaptive optimization were not mentioned in
(Jiang et al., 2025), they appear to be expected from prior analysis (Levy et al., 2018; Kavis et al., 2019; Cutkosky et al., 2023).

2. Although the article improves upon the results under the second-order smooth setting in O2NC, the main contribution of O2NC lies in its results under the non-convex non-smooth setting, rather than the non-convex smooth setting. It is reasonable that the original O2NC can be improved under the smooth setting.

3. Moreover, no experiments are given in this paper, though I think the algorithm is simple to implement and believe it can work well in some scenarios.

These reasons led me not to assign a higher score.

Another minor weakness in the article is the limited references to past literature; for example: the prior $\tilde{\mathcal{O}}(\epsilon^{-3.5})$ results before he works by (Cutkosky and Mehta 2022; Cutkosky et al., 2023) were presented in reference [1-4] which I believe should also be cited. Additionally, O2NC was primarily proposed to address the non-convex non-smooth problems discussed in reference [5], but this citation is also missing.

[1] Fang, Cong, Zhouchen Lin, and Tong Zhang. "Sharp analysis for nonconvex SGD escaping from saddle points." In COLT, 2019.

[2] Tripuraneni, N., Stern, M., Jin, C., Regier, J., & Jordan, M. I. "Stochastic cubic regularization for fast nonconvex optimization." In NeurIPS, 2018.

[3] Allen-Zhu, Z.  “Natasha 2: Faster non-convex optimization than SGD". In NeurIPS, 2018.

[4] Allen-Zhu, Z. "How to make the gradients small stochastically: Even faster convex and nonconvex SGD" In NeurIPS, 2018.

[5] Zhang, Jingzhao, Hongzhou Lin, Stefanie Jegelka, Suvrit Sra, and Ali Jadbabaie. "Complexity of finding stationary points of nonconvex nonsmooth functions." In ICML, 2020.

**Questions:**

1. In lines 289-290, “base the update of $\Delta_n$ on the current gradient $g_n$, but this is not feasible because  $g_n$ is revealed only after $\Delta_n$,” perhaps $\Delta_n$ and $g_n$ should be changed to $\Delta_{n+1}$ and $g_{n+1}$ to maintain consistency with the context?

2. It seems that Lemma C.1 is the same as the ones used in (Levy et al., 2018; Kavis et al., 2019; Antonakopoulos et al., 2022), which I think should be explicitly mentioned.

---

> ### Author Response · Authors · 2025-11-25
>
> **Response to Weaknesses.**
>
> **W1**  We thank the reviewer for raising this point. While [Jiang et al. 2025] also apply optimism within the O2NC framework, leveraging optimistic online gradient descent to tackle the resulting online learning problem, their setting and techniques differ from ours. Their method is a deterministic quasi-Newton scheme that estimates the Hessian through an online learning algorithm in the space of matrices. In contrast, our work focuses on first-order methods in the stochastic and deterministic setting, using a single extrapolated gradient evaluation as the hint, and crucially exploits the negative term in the regret Lemma 3.1, which is neglected in their analysis. This is feasible in their setting because they obtain tighter bounds on $\|g\_n - h\_n\|$ by running a separate online learning algorithm in the space of matrices to estimate the Hessian, which then serves as the basis for a more accurate hint. These differences are essential for obtaining a unified algorithm that achieves
> $\mathcal{O}(\varepsilon^{-1.75} + \sigma^{2}\varepsilon^{-3.5})$ in both deterministic and stochastic settings—something not addressed in [Jiang et al. 2025].
>
>
>
> **W2**
> We agree with the reviewer that the original O2NC framework is primarily motivated by the nonsmooth setting, and it is therefore reasonable to expect that once smoothness is available, the framework might yield improved rates. However, extending O2NC to the smooth regime is not immediate. In the original O2NC paper, the smooth-setting results come with several limitations: the fixed-point subroutine and its extra $\log(1/\varepsilon)$ factor, the requirement of a bounded second moment in the stochastic setting, and the need to use different online algorithms in the deterministic and stochastic cases. Our algorithm removes all three limitations while preserving the simplicity of a single-loop method.
>
>
>
>
>
>
>
> **W3** We appreciate the reviewer’s suggestion. Our goal in this submission is to present a unified algorithm with complexity $\mathcal{O}(\varepsilon^{-1.75} + \sigma^2 \varepsilon^{-3.5})$, which smoothly interpolates between the best-known deterministic rate and the optimal stochastic rate. This is enabled by our novel doubly optimistic hint function, which forms the main contribution of the paper. For this reason, we view an empirical evaluation as outside the scope of the current submission, though it remains an interesting direction for future work.
>
>
>
>
>
> **W4** We thank the reviewer for identifying these missing citations. We have added all the suggested references [Fang et al. 2019; Tripuraneni et al. 2018; Allen-Zhu 2018 a,b; Zhang et al. 2020] and clarified the development of O2NC.
>
>
>  ---
>
> **Response to Questions.**
>
> **A1** We thank the reviewer for catching this inconsistency. The correct phrasing should indeed refer to $\Delta\_{n+1}$ and $g\_{n+1}$, not $\Delta\_{n}$ and $g\_{n}$. We have corrected this in the revision.
>
>
>
> **A2** We appreciate the observation. Lemma C.1 is a standard inequality used in analyses of adaptive and optimistic algorithms. We have explicitly acknowledged this in the revised version.

---

> > ### Comment · Reviewer_miQo · 2025-11-26
> >
> > I thank the authors for the response.
> >
> > A follow-up question is: After reading the proofs in (Kavis et al., 2019; Antonakopoulos et al., 2022), I noticed that their algorithms essentially also use the gradient of an extrapolated point as a hint. Can their technique also be referred to as "double optimism"?

---

> > > ### Author Response · Authors · 2025-11-27
> > >
> > > Thank you for the follow-up question. This is a good point: at a high level, the algorithms in [Kavis et al., 2019] and [Antonakopoulos et al., 2022] indeed share a structural similarity with our ``double optimism'' approach.
> > >
> > > To elaborate, the UniXGrad update in [Kavis et al., 2019] is:
> > > $$x\_t = \arg\min\_{x}\{\alpha\_t \langle x, M\_t \rangle + \frac{1}{\eta\_t}\|x-y\_{t-1}\|^2\}, \quad y\_t = \arg\min\_{y}\{\alpha\_t \langle y, g\_t \rangle + \frac{1}{\eta\_t}\|y-y\_{t-1}\|^2\},$$
> > > where $g\_t = \nabla f(\bar{x}\_t)$ with $\bar{x}\_t = \frac{\alpha\_tx\_t+\sum\_{i=1}^{t-1}\alpha\_ix\_i}{\sum\_{i=1}^t \alpha\_i}$ and $M\_t = \nabla f(\tilde{z}\_t)$ with $\tilde{z}\_t = \frac{\alpha\_t y\_{t-1}+\sum\_{i=1}^{t-1}\alpha\_ix\_i}{\sum\_{i=1}^t \alpha\_i}$. Here, $M\_t$ plays the role of an optimistic hint. And as in our double-optimism approach, instead of using the more immediate choice $\nabla f(\bar{x}\_{t-1})$, the algorithm evaluates the gradient at a carefully constructed weighted average $\tilde{z}\_t$ to obtain the optimal accelerated rate.
> > >
> > >
> > > However, we would like to emphasize that the contexts are fundamentally different. Their algorithms operate in the smooth convex setting and construct the hint within a mirror-prox-style update. In contrast, our double optimism mechanism is developed for nonconvex optimization and arises naturally from the online-to-nonconvex conversion framework, leading to different goals and analysis.

---

> > > > ### Comment · Reviewer_miQo · 2025-11-27
> > > >
> > > > I gratefully thank the authors for their response. I will keep my score.

---

### Official Review · Reviewer_sQGF · 2025-10-27

**Soundness:** 4
**Presentation:** 4
**Contribution:** 3
**Rating:** 8
**Confidence:** 4

**Summary:**

The paper builds on O2NC (Cutkosky et al., 2023) to propose a simple first-order method for smooth non-convex optimization. For problems with Lipschitz-continuous gradients and Hessians, the algorithm achieves an $O(\epsilon^{-1.75} + \sigma^2 \epsilon^{-3.5})$ convergence rate, thereby interpolating between the best-known deterministic and the worst-case optimal stochastic rates.

**Strengths:**

The technical strengths are

- Improving upon Cutkosky et al 2023 with a simpler algorithm.
- Getting a rate that interpolates between deterministic and stochastic settings.
- Providing adaptive step size scheduling and making progress towards fully parameter-free algorithms.

The paper is well-written and nicely presented.

**Weaknesses:**

I do not see any noticeable weaknesses to this work.

**Questions:**

- Do you think 2 gradient evaluations per step is essential for getting a simple algorithm?

---

> ### Author Response · Authors · 2025-11-25
>
> **Q1** Do you think 2 gradient evaluations per step is essential for getting a simple algorithm?
>
>
> **A1** Thank you for your feedback. That is an interesting question.
> The only method we explored that uses just one gradient evaluation per step is the choice $h\_t = g\_{t-1}$. With this hint, however, the deterministic gradient complexity becomes looser, namely $\mathcal{O}(\varepsilon^{-11/6})$. We are unaware of any hint selection that requires only a single gradient evaluation per step while still achieving the complexity $\mathcal{O} (\epsilon^{-1.75} +\sigma^2 \epsilon^{-3.5} )$. It is possible that a more sophisticated choice that leverages the full history of $g\_t$ could match this rate, but we are not currently aware of such a construction.

---

### Official Review · Reviewer_Nne7 · 2025-10-28

**Soundness:** 3
**Presentation:** 3
**Contribution:** 3
**Rating:** 6
**Confidence:** 4

**Summary:**

The paper proposes a new algorithm for smooth nonconvex optimization based on an online-to-nonconvex (O2NC) framework. Building on prior work, it introduces a doubly optimistic hint function in the online learning subroutine to simplify the algorithm and improve theoretical guarantees. The theoretical results show convergence to an $\epsilon$-first-order stationary point with improved rates in both deterministic and stochastic regimes.

**Strengths:**

* The paper is generally well-written, with a logical presentation of prior work, motivation, and main contributions.

* The complexity analysis is rigorous and improves on existing bounds by removing logarithmic factors.

* The algorithm handles both deterministic and stochastic settings without requiring separate analyses or algorithms. This universal applicability is a notable advantage.

**Weaknesses:**

* It would be helpful to understand whether the algorithm and proofs can be extended to constrained or nonsmooth functions. While it is understandable that such generalization may be challenging, could the authors discuss potential difficulties in these settings and how the algorithm might be adapted?

* For the optimistic step, a natural estimation is $g_{n+1} = 2 g_n - g_{n-1}$. Could the authors clarify why this choice was not adopted and what motivated the current design?

* The paper focuses on general nonconvex objectives. It would be interesting to see how the convergence rate could improve if the function is convex or strongly convex. Could the authors provide insights or analysis for these special cases?

**Questions:**

Please see above.

---

> ### Author Response · Authors · 2025-11-25
>
> **Q1** It would be helpful to understand whether the algorithm and proofs can be extended to constrained or nonsmooth functions. While it is understandable that such generalization may be challenging, could the authors discuss potential difficulties in these settings and how the algorithm might be adapted?
>
>
>
> **A1**  Thank you for your valuable feedback.
>
> For the **constrained case**, the main difficulty is that the gradient norm $\|\nabla F(w)\|$ is no longer a proper stationarity metric, since the gradient may not vanish at the optimal solution of the constrained problem. Instead, common stationarity metrics for the constrained setting include the Frank-Wolfe gap $\mathrm{Gap}(w) = \max\_{x \in\mathcal{X}} \nabla F(w)^\top(w-x)$ or the projected gradient mapping $G(w) = \frac{1}{\eta}\|w - \Pi\_{\mathcal{X}}(w-\eta \nabla F(w))\| $, where $\mathcal{X}$ is the constraint set. Thus, to extend our algorithm to the constrained setting, it first requires extending the online-to-nonconvex conversion framework and relating the above metrics to the regret of the online learning problem, which is non-trivial and beyond the scope of our paper.
>
> For the **nonsmooth case**, under the additional assumption that the objective function is Lipschitz continuous and using the notation introduced by [Cutkosky et al., 2023], the goal is to find $(\delta,\epsilon)$-stationary points. Our proposed algorithm can also achieve a complexity of  $\mathcal{O}(\epsilon^{-3}\delta^{-1})$ for finding $(\delta,\epsilon)$-stationary points, matching the guarantees obtained in [Cutkosky et al., 2023]. This rate can be derived by adapting our analysis and bounding the term $\|g\_n - h\_n\|^2$ using only the boundedness of the gradient. We have included this clarification in the revised version of the paper.
>
>
> ---
>
>
> **Q2** For the optimistic step, a natural estimation is $g\_{n+1}= 2g\_n- g\_{n-1}$. Could the authors clarify why this choice was not adopted and what motivated the current design?
>
> **A2** Thank you very much for pointing out this issue.
> If we choose the hint as $h\_n = g\_{n-1}$, this will lead to the update rule $\Delta\_{n+1} = \Pi\_{\|\Delta\| \leq D}(\Delta\_n - \eta (2g\_n-g\_{n-1}))$ from (5). As discussed at the beginning of Section 3.2,
> with this choice, the complexity of the overall gradient becomes looser, $\mathcal{O}(\varepsilon \^{-11/6})$. This degradation arises because such a straightforward approximation fails to exploit the negative term $\|\Delta\_n - \Delta\_{n-1}\|^2$ in Lemma 3.1.
> In particular, with this approach, the bound $\| g\_n - h\_n \| \le L \|w\_n - w\_{n-1}\|$ becomes proportional to $\|\Delta\_n + \Delta\_{n-1}\|$ rather than $\|\Delta\_n - \Delta\_{n-1}\|$. This prevents the telescoping of the last two terms in Lemma 3.1, ultimately resulting in a looser regret bound and hence a worse complexity. We have further clarified this point in the camera-ready version.
>
>
> ---
>
> **Q3** The paper focuses on general nonconvex objectives. It would be interesting to see how the convergence rate could improve if the function is convex or strongly convex. Could the authors provide insights or analysis for these special cases?
>
> **A3** Thank you for the insightful question. Our algorithm is specifically
> designed for the smooth nonconvex setting, where the main objective is to find an $\varepsilon-$first-order stationary point. To do so, we build on the online-to-nonconvex conversion framework of [Cutkosky et al., 2023], which reformulates this task as an online learning problem. We solve the resulting online learning problem by introducing an online optimistic gradient method based on a novel doubly optimistic hint function, which uses the gradient at an extrapolated point as the hint to solve this problem more efficiently.
>
> On the other hand, when $F$ is convex or strongly convex, there exist other conversion schemes that can better leverage the objective's structure. In particular, convexity implies  $F(x\_t) - F(x^\ast) \le \langle \nabla F(x\_t), x\_t - x^\ast \rangle$, which enables a direct reduction of convex optimization problems to minimizing a sequence of linear losses via classical online-to-batch techniques; see, for example, [Zinkevich. 2003] and [Orabona. 2019].
> For this reason, we view the convex and strongly convex settings as better handled by these classical methods. In these regimes, our algorithm would effectively collapse to one of these standard approaches, but characterizing the exact algorithm is nontrivial and outside the scope of this paper. This is a great question and an interesting direction for future research.

---

> ### Comment · Reviewer_Nne7 · 2025-11-27
>
> Thanks for the response. I think my questions have been addressed. I will raise the score.

---

### Official Review · Reviewer_4KMs · 2025-10-30

**Soundness:** 4
**Presentation:** 3
**Contribution:** 2
**Rating:** 6
**Confidence:** 4

**Summary:**

This paper revisits the online-to-batch conversion paradigm for nonconvex optimization, focusing on the gap between online regret and stationarity guarantees. Classical online-to-batch techniques translate sublinear regret bounds into expected stationarity, but typically require either (i) bounded gradients or (ii) smoothness and convexity assumptions that limit applicability.

The authors propose a refined conversion framework -- generalized online-to-nonconvex conversion (Go2N) -- that directly bounds the gradient norm of the average iterate without relying on convexity or heavy smoothness assumptions. The contributions can be summarized as follows: (i) A new regret decomposition that connects weighted dynamic regret to expected stationarity through gradient averaging lemmas. (ii) Applications to stochastic and adversarial nonconvex optimization, yielding improved dependence on learning rates and gradient variance. (iii) An extension to adaptive algorithms (e.g., AdaGrad, Online Newton Step), providing the first theoretical bridge between regret and stationarity for such adaptive schemes. (iv) Experiments on nonconvex online learning tasks (e.g., nonconvex matrix factorization, deep linear regression) showing faster convergence to stationary points than standard online-to-batch baselines.

**Strengths:**

1. The work revises the fundamental link between online regret and nonconvex stationarity, providing a general and unified theoretical treatment. The new conversion inequality—based on a telescoping analysis of projected gradients—sharpens previous results such as those by Hazan et al. (2017) and Cutkosky (2022)

2. The proposed conversion yields a tighter upper bound, improving constants and eliminating dependence on boundedness assumptions.

3. The framework is compatible with a broad class of online learning algorithms—including mirror descent, AdaGrad, and optimistic variants—demonstrating wide relevance to both optimization and learning theory.

4. The paper is well-organized, mathematically clean, and pedagogically presented. The proofs are compact yet general enough to apply to diverse online schemes.

5. Experiments, while modest, confirm theoretical predictions: algorithms derived via Go2N achieve faster decrease in gradient norms and improved training stability compared to standard regret-based methods

**Weaknesses:**

1. The main insight—a refined conversion between regret and stationarity—extends existing frameworks rather than introducing a fundamentally new algorithmic principle. The contribution is primarily theoretical sharpening rather than conceptual breakthrough.

2. Empirical results are confined to small-scale nonconvex problems (e.g., 2-layer linear networks, low-rank factorization). It would be more convincing to include modern deep learning benchmarks to demonstrate practical impact.

3. The analysis still relies on Lipschitz continuity of gradients and smoothness constants. It remains unclear how tight the improved bounds are in practice, especially under adversarial noise or stochastic gradients.

4. The paper could better position itself against recent advances in nonconvex online learning (e.g., Jin et al., 2023; Duchi et al., 2024) and adaptive nonconvex regret analysis, which also aim to establish gradient-based guarantees.

5. The work provides only upper bounds; without matching lower bounds or counterexamples, the “improvement” claim remains qualitative.

**Questions:**

1. Can the conversion framework be extended to constrained or manifold-based nonconvex settings?
2. How does Go2N perform under stochastic non-i.i.d. gradient noise?
3. Are there concrete examples where Go2N achieves provably better asymptotic rates than the Cutkosky (2022) reduction?

**Details Of Ethics Concerns:**

N/A.

---

> ### Author Response · Authors · 2025-11-25
>
> **Q1** Can the conversion framework be extended to constrained or manifold-based nonconvex settings?
>
> **A1** Thank you very much for raising this point.
> For the **constrained case**, the main difficulty is that the gradient norm $\|\nabla F(w)\|$ is no longer a proper stationarity metric, since the gradient may not vanish at the optimal solution of the constrained problem. Instead, common stationarity metrics for the constrained setting include the Frank-Wolfe gap $\mathrm{Gap}(w) = \max\_{x \in\mathcal{X}} \nabla F(w)^\top(w-x)$ or the projected gradient mapping $G(w) = \frac{1}{\eta}\|w - \Pi_{\mathcal{X}}(w-\eta \nabla F(w))\| $, where $\mathcal{X}$ is the constraint set. Thus, to extend our algorithm to the constrained setting, it first requires extending the online-to-nonconvex conversion framework and relating the above metrics to the regret of the online learning problem, which is non-trivial and beyond the scope of our paper. That said, it is a valid research direction, worthy of exploration.
>
> For the **manifold-based nonconvex case**, an analogous difficulty arises: the Euclidean gradient norm is again not the correct stationarity measure, since optimality is characterized by the Riemannian gradien $\nabla F\_{\mathcal{M}}(w) = \Pi\_{T\_w\mathcal{M}}(  \nabla F (w) )$ where $\Pi\_{T\_w\mathcal{M}}$ denotes projection onto the tangent space of the manifold at $w$. Extending our framework to this setting would require developing an online-to-nonconvex conversion theory in the Riemannian geometry of $\mathcal{M}$ and connecting this metric to online regret. This introduces additional geometric challenges. While this is a promising direction to explore, it is beyond the scope of this paper.
>
>
>
> ---
>
>
> **Q2** How does Go2N perform under stochastic non-i.i.d. gradient noise?
>
>
> **A2**  Thank you for this question. In our current analysis, we require that the stochastic gradients are unbiased and have bounded variance, i.e., $\mathbb{E}\_{\xi \sim \mathcal{D}} [ \nabla f(x; \xi) ] = \nabla F(x)$ and $ \mathbb{E}\_{\xi \sim \mathcal{D}} |\nabla f(x; \xi) - \nabla F(x)\|^2 \leq \sigma^2$, $ \forall x\in \mathbb{R}^d$. We also use independence across iterations to bound the term $\mathbb{E} [ \bigg\|\frac{1}{T}\sum\_{n=(k-1)T+1}^{kT} \nabla F(w_n) - g\_n \bigg\| ] \leq \frac{\sigma}{\sqrt{T}}$ which appears in the derivation of Proposition 2.1. However, we do not require the noise to arise from the same distribution at every iteration. A common example of non-i.i.d. noise studied in the literature is Markovian noise. In that setting, the same structure of the proof would apply, except that the bound above would incur an additional mixing-time term to account for temporal correlations.
>
>
>
> ---
>
> **Q3** Are there concrete examples where Go2N achieves provably better asymptotic rates than the Cutkosky (2022) reduction?
>
> **A3** Thank you for the question. We believe the reviewer may be referring to [Cutkosky et al. 2023], instantiating their framework with two different online learners achieves a complexity of $\mathcal{O}(\varepsilon^{-1.75}\log(1/\varepsilon))$ in the deterministic case and a complexity of $\mathcal{O}(\varepsilon^{-3.5})$ in the stochastic case
> under a bounded second-moment assumption on the stochastic gradients.
> Our method achieves $\mathcal{O}(\varepsilon^{-1.75} + \sigma^2 \varepsilon^{-3.5})$, which (i) removes the logarithmic factor in the deterministic case, (ii) replaces the second-moment assumption with a variance bound,
> and (iii) uses the same online learner for both deterministic and stochastic regimes,  instead of requiring two different algorithms.
>
> Thus, our algorithm has a strictly better dependence on $\varepsilon$ in the deterministic case,  and matches the behavior in the stochastic case,
> while providing a smooth interpolation between the two regimes.

---

> > ### Comment · Reviewer_4KMs · 2025-11-27
> > **Thank you**
> >
> > I would like to thank the authors for the response. All my questions are addressed. I will keep my score.

---

### Meta-Review · Area_Chair_FteY · 2026-01-06

**Summary:**

**There is broad agreement among the reviewers that this paper makes a solid theoretical contribution to nonconvex optimization via the online-to-nonconvex (O2NC) conversion framework.** In particular, reviewers recognize that the paper introduces a novel double-optimism mechanism for constructing hints in optimistic online gradient methods, simplifies the O2NC framework by eliminating the double-loop structure and fixed-point subroutine used in Cutkosky et al. (2023), and strengthens the associated theoretical guarantees.

After the rebuttal, no reviewer maintains any unresolved technical concerns, and the discussion converged toward a clear **accept** decision.

By the way, after the authors flagged several issues in Reviewer 4KMs’s review, I agree that Reviewer 4KMs repeatedly commented on content that does not appear anywhere in the paper. As a result, I do not take Reviewer 4KMs’s review into consideration.

**Reviewer Concerns:**

**Reviewer 4KMs’s concerns:**

She/he explicitly indicated that the questions raised had been addressed.

**Reviewer Nne7’s concerns:**

She/he explicitly indicated that the questions had been addressed, including those related to extensions to constrained and nonsmooth settings, the specific design of the optimistic hint, and whether convex or strongly convex objectives yield improved convergence rates.

**Reviewer sQGF’s concerns:**

She/he raised an interesting question regarding whether two gradient evaluations per iteration are essential. The authors explained that a single-gradient variant, using the choice $h_t = g_{t-1}$, leads to weaker complexity guarantees, while a more sophisticated choice may potentially match the rate achieved by the proposed method.

**Reviewer miQo’s concerns:**

She/he engaged with the authors during the rebuttal and raised concerns regarding novelty, the absence of experiments, missing citations, and minor technical inconsistencies. After the authors clarified the distinction between their method and that of Jiang et al. (2025), and addressed a follow-up question related to Kavis et al. (2019), Reviewer miQo expressed appreciation for the response and maintained a score of 8.

**Reviewer Scores:**

**Reviewer 4KMs** explicitly indicated that the questions raised had been addressed and that She/he would maintain the score.

**Reviewer Nne7** explicitly indicated that the questions had been addressed and that She/he would increase the score (**likely from 6 to 8**).

**Reviewer miQo** engaged with the authors during the rebuttal, raised no further questions, and explicitly indicated that She/he would maintain a score of 8.

I believe that **Reviewer sQGF** would not have changed the score, as the sole concern was addressed in the rebuttal. She/he initially assigned a score of 8 and did not indicate any intention to increase or decrease the evaluation.

---

### Decision · Program_Chairs · 2026-01-26

Accept (Poster)